# Genetic architecture of routinely acquired blood tests in a British South Asian cohort

Benjamin M. Jacobs [1,2,20], Daniel Stow [1,20], Sam Hodgson [1,20], Julia Zöllner [1,3,20], Miriam Samuel [1,20], Stavroula Kanoni [4], Saeed Bidi [1], Genes & Health Research Team*, Klaudia Walter [5], Claudia Langenberg [6,7], Ruth Dobson [1,2], Sarah Finer [1,8], Caroline Morton[1], Moneeza K. Siddiqui [1,21], Hilary C. Martin [5,21], Maik Pietzner [6,7,21], Rohini Mathur[1,21] & David A. van Heel [1,8,21] ✉

Understanding the genetic basis of routinely-acquired blood tests can provide insights into several aspects of human physiology. We report a genome-wide association study of 42 quantitative blood test traits defined using Electronic Healthcare Records (EHRs) of ~50,000 British Bangladeshi and British Pakistani adults. We demonstrate a causal variant within the *PIEZO1* locus which was associated with alterations in red cell traits and glycated haemoglobin. Conditional analysis and within-ancestry fine mapping confirmed that this signal is driven by a missense variant - chr16-88716656-G-$T_T$ - which is common in South Asian ancestries (MAF 3.9%) but ultra-rare in other ancestries. Carriers of the T allele had lower mean HbA1c values, lower HbA1c values for a given level of random or fasting glucose, and delayed diagnosis of Type 2 Diabetes Mellitus. Our results shed light on the genetic basis of clinically-relevant traits in an under-represented population, and emphasise the importance of ancestral diversity in genetic studies.

Studying the genetic basis of complex traits and diseases across ancestries has the potential to improve fine-mapping resolution, uncover novel biology, and ensure that downstream applications, such as polygenic risk score profiling, perform equitably across populations[1–6]. Although there is increasing ancestral diversity in published genome-wide association studies (GWAS)[7,8], our understanding of the genetic basis of complex traits and diseases remains skewed towards populations of European ancestral backgrounds[9]. Participants of South Asian ancestry, despite representing ~25% of the world's population, make up only 2% of published GWAS, even in

relatively diverse cohorts such as UK Biobank (UKB) (~2%) and the Million Veterans Programme (<1%)[7–9].

British South Asians constitute around 9% of the UK population (UK census 2021, https://www.ons.gov.uk/), but around 20% of the UK population with diabetes[10]. Both the prevalence of diabetes and the rates of diabetic complications are higher among British South Asians. In addition British South Asians have higher rates of cardiovascular disease (CVD), tend to develop CVD at a younger age, and are at higher risk of early adverse outcomes such as myocardial infarction and stroke[10]. The differences in allele frequency and linkage disequilibrium between this

[1]Wolfson Institute of Population Health, Queen Mary University of London, London, UK. [2]Department of Neurology, Royal London Hospital, Barts Health NHS Trust, London, UK. [3]University College London, London, UK. [4]William Harvey Research Institute, Queen Mary University of London, London, UK. [5]Wellcome Sanger Institute, Wellcome Genome Campus, Hinxton, UK. [6]Precision Healthcare University Research Institute, Queen Mary University of London, London, UK. [7]Computational Medicine, Berlin Institute of Health at Charité – Universitätsmedizin Berlin, Berlin, Germany. [8]Blizard Institute, Queen Mary University of London, London, UK. [20]These authors contributed equally: Benjamin M. Jacobs, Daniel Stow, Sam Hodgson, Julia Zöllner, Miriam Samuel. [21]These authors jointly supervised this work: Moneeza K Siddiqui, Hilary C. Martin, Maik Pietzner, Rohini Mathur, David A van Heel. *A list of authors and their affiliations appears at the end of the paper. ✉e-mail: d.vanheel@qmul.ac.uk

population and populations of European ancestry can help to power the discovery of novel variants associated with traits and diseases which may be missed in Eurocentric analyses[11]. The genetic architecture of this cohort—which is enriched for consanguinity compared with other large cohorts such as UKB—provides a unique opportunity for understanding the contribution of rare, deleterious coding variation and homozygosity to complex traits and diseases[12–14]. Understanding the genetic basis of variation in common blood tests (such as glycaemic markers and lipids) in cohorts of South Asian ancestry may therefore help to explain some of these stark healthcare disparities and may shed light on biology of relevance to people of all ancestries[3,11,15].

Here we report the largest genome-wide association study in a cohort of South Asian ancestry for a range of routinely acquired blood test parameters using data from the Genes & Health cohort—a longitudinal genotype-phenotype study comprising ~50,000 British individuals of South Asian ancestry (British Bangladeshi and British Pakistani) recruited in the United Kingdom. Through multi-ancestry meta-analysis and fine mapping, we discover causal genetic signals which vary in impact across ancestries, demonstrating the power of diversity in genetic analysis of complex traits.

## Results

We tested the association of ~4.8 million common imputed genetic variants (MAF > 0.01, INFO > 0.7) with 42 routinely acquired quantitative blood tests derived from Electronic Healthcare Records (EHR) in the Genes and Health cohort—a genotype-phenotype cohort comprising ~50,000 British individuals of South Asian ancestry (Supplementary Data 1 and 2). The number of participants included in the GWAS varied by phenotype, ranging from 13,870 (AST: aspartate aminotransferase) to 38,224 (Haemoglobin, Supplementary Data 3, Fig. 1A). Across all 42 traits we found 517 study-wide significant independent locus-trait associations in the Genes and Health cohort

(Fig. 1A, $P < 1.2 \times 10^{-9} = 5 \times 10^{-8}/42$; Supplementary Data 3 and 4). There was no substantial inflation of test statistics (median $\lambda_{GC}$ 1.07, range 1.0–1.11). Linkage disequilibrium score regression (LDSC) intercepts ranged from 0.95 (Platelets) to 1.02 (Vitamin B12, Fig. 1B, Supplementary Data 3), falling within the range reported for these traits in UKB and confirming the absence of significant population stratification or other systematic bias[16].

We replicated several well-known associations across all traits tested, such as for LDL-cholesterol (*PCSK9, APOB, LDLR, HMCGR,* and *APOE*[2]), vitamin D (*GC*)[17], creatinine (*GATM*)[18], and HbA1c (*HHEX, GCK,* and *TCF7L2;* Fig. 1C)[6]. We detected some associations where the SNP was common (MAF ≥ 0.01) in people of South Asian ancestry but ultrarare (MAF < 0.0001) in other ancestral populations, and therefore likely to be missed in genetic analysis restricted to European ancestry. For example, we replicated the previously reported associations between missense variants in *LIPG* with HDL-C[2], in *IL7*[11] with lymphocyte count, and in *GPI1B*[19] with platelet count (Supplementary Data 4) which reflect signals common in South Asian populations but rare in other ancestries. For the 29 traits which overlapped with UKB, all of the detected signals in the Genes & Health cohort were within regions harbouring suggestive associations in UKB (i.e. within ±1MB of a variant with $P < 1 \times 10^{-5}$ in at least one ancestral group in UKB). The genetic architecture of all traits was largely concordant between the Genes & Health cohort—which is primarily of Bangladeshi or Pakistani genetic ancestry—and European-ancestry UKB participants (cross-ancestry genetic correlations all >0.35, Fig. 2, Supplementary Data 5), however for some traits—such as HbA1c, lymphocyte count and random glucose—the correlation was significantly lower than 1 ($\alpha = 5\%$, $P < 0.05/29 = 0.001$), indicating heterogeneity in the genetic architecture of these traits across ancestries.

Through multi-ancestry meta-analysis with UKB (Supplementary Data 6), adjusted for test statistic inflation at the per-cohort level, we

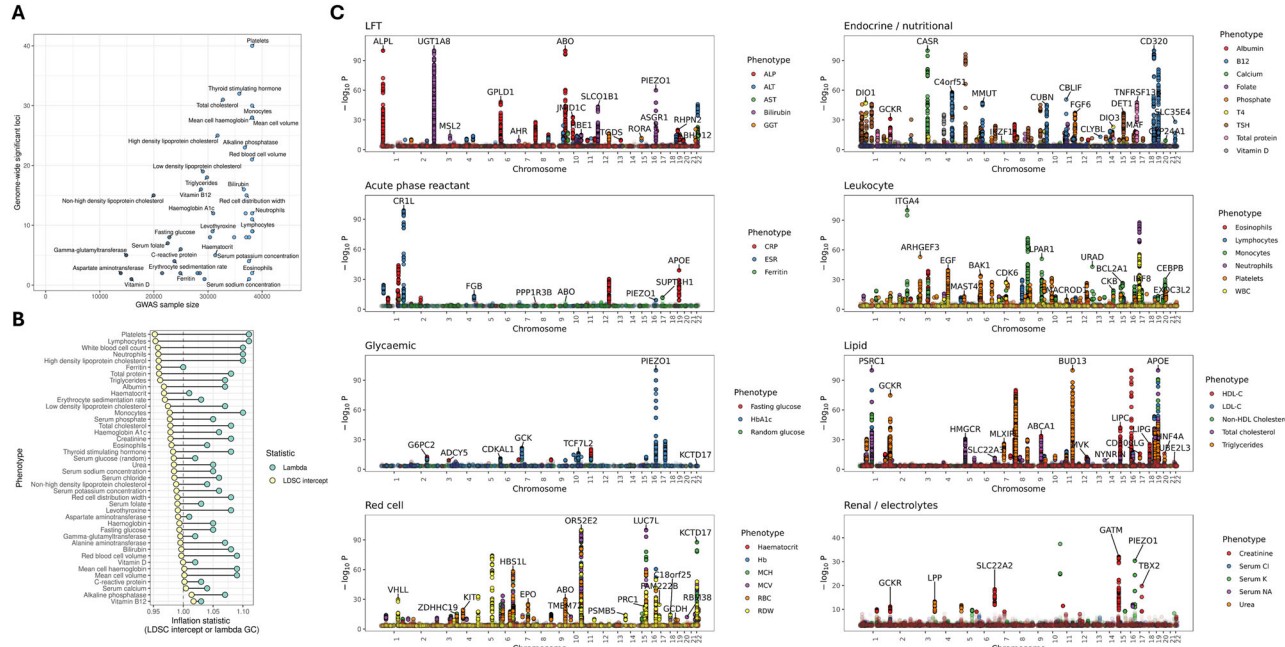

**Fig. 1 | GWAS of routinely acquired blood tests in a South Asian ancestry cohort. A** Scatter plot displaying the relationship between GWAS sample size (x axis) and the number of study-wide significant independent genetic loci for each trait examined in the Genes & Health Cohort. Independent loci were defined using in-sample LD to clump results (distance 1KB, $R^2$ 0.001) and a study-wide significance threshold of $P < 1.2 \times 10^{-9}$. **B** Forest plot indicating the degree of test statistic inflation for each GWAS trait. The *x* axis shows the inflation statistic—either the LDSC intercept (yellow) or the genomic inflation factor ($\lambda$, light green). The *y*-axis

indicates the trait. **C** Manhattan plots showing the results from the Genes & Health GWAS. *P* values are truncated at $P < 1e-100$ for clarity. Traits are divided by broad subtype. For each trait category, only SNPs passing the study-wide significance threshold are shaded in full colour. Other SNPs are translucent. SNPs with $P > 0.001$ are not shown for clarity. The top SNP (i.e. lowest *P* value) within each category for each chromosome is annotated with the nearest gene. Some labels are omitted for clarity. *P* values reflect the output of the GWAS models (mixed linear models) implemented in REGENIE.

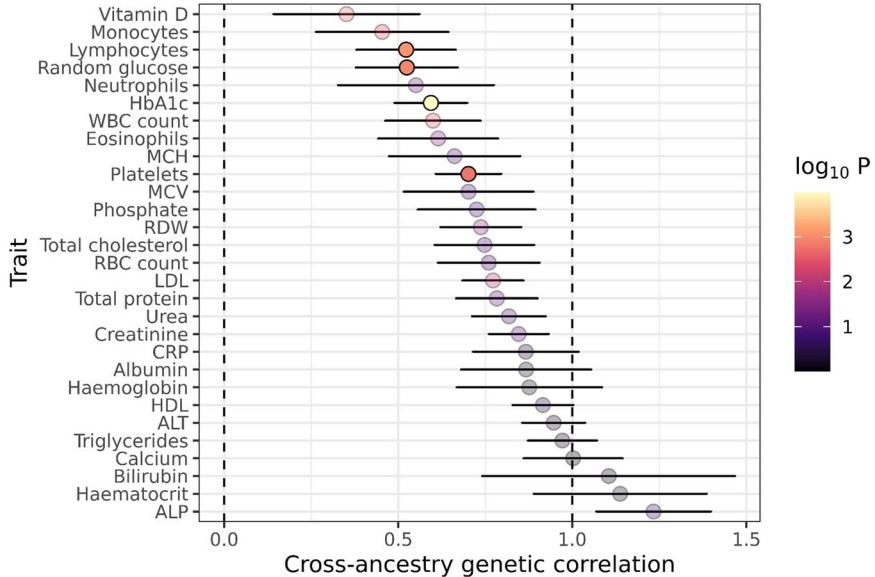

**Fig. 2 | Cross-ancestry genetic correlations between South Asian ancestry GWAS in the Genes & Health cohort and UK Biobank European-ancestry GWAS.** Forest plot showing the cross-ancestry genetic correlation (genetic effect) estimated for each trait between the Genes & Health GWAS and UK Biobank European-ancestry GWAS. The points are shaded by the −log10(P) for the P value assessing the hypothesis that the estimate is below one (Z-test). The dashed lines indicate 0 i.e., no genetic correlation, and 1 i.e., perfect genetic correlation. Horizontal bars indicate the 95% confidence intervals. The centre of the bars reflects the estimate for the cross-ancestry genetic correlation. Note that the estimate is unbounded, hence in some cases the estimate is over one. Points are greyed out if the estimate was not significantly different from 1 at a study-wide significance threshold of $P < 0.05/29$. Sample sizes used for these estimates are given in Supplementary Data 3 (for Genes & Health) and Supplementary data 6 (for UK Biobank).

identified 153 locus-trait associations which achieved genome-wide significance and varied in magnitude according to ancestry ($P_{Meta-analysis} < 1.72 \times 10^{-9}$ and $P_{Heterogeneity} < 1.72 \times 10^{-9}$, Supplementary Data 7, Fig. 3)[20]. These regions of ancestral heterogeneity implicated genes where genetic variants with differential allele frequences across ancestries may have a sizeable impact on blood test values. For example, HbA1c is a marker of long-term blood glucose control which measures the concentration of glycated haemoglobin molecules inside red blood cells. Genetic variants that influence red blood cell morphology and survival—such as those that cause Sickle Cell Disease—can therefore modify HbA1c, and are often highly variable in terms of allele frequency between populations. Our meta-analysis identified several genes associated with Mendelian haemoglobinopathies as sites of ancestral heterogeneity in the genetic architecture of HbA1c (e.g. spectrin [*SPTA1*], ankyrin [*ANK*], *PIEZO1*).

To determine whether these sites of cross-ancestry heterogeneity were explained by causal variants with varying impact across ancestries (e.g. due to a large difference in allele frequency), we performed multi-ancestry fine mapping by combining these GWAS results with within-ancestry GWAS from the major ancestral populations in UKB. We discovered several examples of variants associated with quantitative blood test traits which were common in South Asian populations but rare or ultra-rare in UKB participants of European, African, or East Asian ancestry (Supplementary Data 8). For instance, we fine mapped a causal signal for HDL cholesterol (HDL-C) to a single variant—the intronic variant chr18-49568194-T-C (rs576653339, also reported in ref. 2), which has a frequency of 2.2% in South Asian populations but is ultra-rare in other populations (gnomAD frequencies 0.01% in EAS, 0.001% in NFE). This is an intronic variant in *LIPG*, the gene encoding endothelial lipase, in which loss-of-function variants are known to impact HDL cholesterol[21]. We also fine-mapped several variants within the extended *HBB* locus (chr11:4686341-11:5905046 on hg38) which appeared to be causal variants for a range of red blood cell traits (Supplementary Data 8) and were appeared to exert SAS-specific effects despite being common in other ancestries. For example, the

variant chr11-5026936-T-A (rs113322016), an intergenic SNV located ~200KB from the beta-globin gene *HBB*, was associated with both RDW and RBC mass, with the A allele increasing both. The minor allele (chr11-5026936-T-A$_A$) is common in SAS (MAF 21.7%) and AFR (19.4%) 1000 genomes samples, and relatively rarer in EUR samples (MAF 4.7%). The intronic variant chr16-205386-G-A (located within *LUC7L* and within ~30KB of the alpha haemoglobin genes *HBA1* and *HBA2*) was identified as a causal variant for a range of red cell traits in both SAS cohorts only, in keeping with its ultra-rare frequency in non-SAS populations—although the effect has been previously reported in UKB SAS individuals, we add confidence to this result by replicating in a far larger cohort of people of SAS ancestry[11]. Given the complex LD and structural variation at the globin loci, further work with sequencing is required to resolve whether these variants are truly causal or are tagging structural variants.

Fine mapping of the *PIEZO1* association with HbA1c (chr16:88416016-89416016) yielded six distinct causal signals, of which five were mapped to single variant resolution (Supplementary Data 8, Fig. 4). One of these six causal signals (Credible Set 1 in Fig. 4B, fine mapped to chr16-88716656-G-T) only showed evidence of causality in the cohorts of SAS ancestry, i.e. the post-hoc probability of being causal was >0.8 in both Genes & Health and SAS-ancestry UKB, but not in other ancestral groups. Variation at *PIEZO1* has been previously associated with HbA1c, and is one of several loci which influence HbA1c via an effect on red cell lifespan rather than glycaemic control[22]. The post-hoc causal probability of chr16-88716656-G-T was 1.0 in both Genes & Health and UKB SAS samples respectively. The allele frequency of this variant was low (MAF < 0.01) in the other populations tested (in UKB EUR, AFR and EAS-ancestry participants), in keeping with the reported frequency of this variant in gnomAD (3.9% in SAS, and <0.01% in Non-Finnish Europeans [NFE], African, and East Asian ancestral groups). A second variant, chr16-88524059-C-T, showed strong evidence of being causal (posterior probability 0.99) in Genes & Health, low probability in EUR, EAS, and AFR UKB samples, and moderate evidence (posterior probability 0.74) in UKB SAS-ancestry

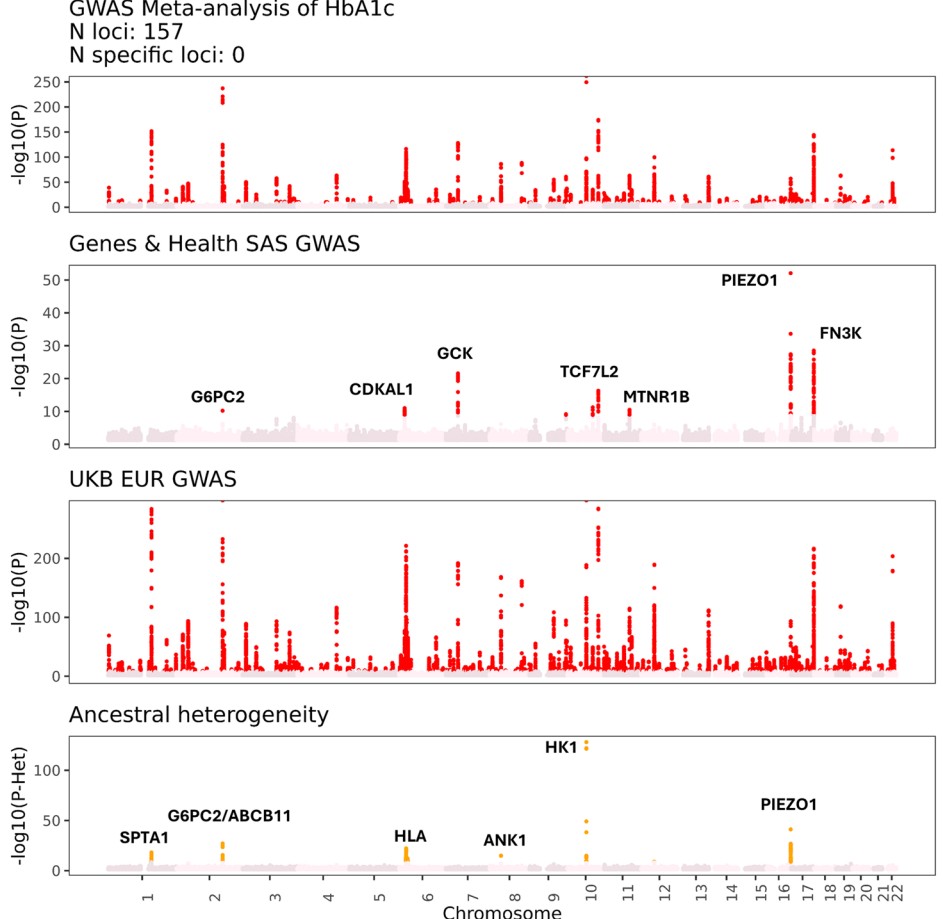

**Fig. 3 | Multi-ancestry meta-analysis of glycated Haemoglobin (HbA1c).** Manhattan plots showing the GWAS of HbA1c in the multi-ancestry meta-analysis (top panel), the Genes & Health GWAS (second panel), the UKB EUR GWAS (third panel), and the P values for ancestral heterogeneity (bottom panel) for each SNP derived from the meta-analysis. Selected genes are highlighted. Multi-ancestry meta-analysis was conducted using MR-MEGA. Ancestral heterogeneity P values refer to the strength of evidence supporting the hypothesis that SNP effect sizes at a locus were correlated with ancestral principal components, derived from summary statistics.

samples. This is an intronic SNP lying between exons four and five of *ZPFM1*, which is common in gnomAD SAS samples (MAF 4.3%) but also relatively common in gnomAD NFE samples (MAF 1.9%).

To validate these findings from multi-ancestry fine mapping we conducted single-ancestry fine mapping within Genes & Health. This analysis recapitulated the finding that the *PIEZO1* missense variant (chr16-88716656-G-T, $\beta = -0.53$, MAF 3.9%, $P = 2.0 \times 10^{-147}$) is the most likely causal variant at the locus (PIP 1) under the single causal variant assumption (Supplementary Data 9). Conditional analysis confirmed that chr16-88716656-G-T accounted for the entirety of the signal at this locus, as no other signals persisted at study-wide significance ($P < 1.72 \times 10^{-9}$) on adjustment for chr16-88716656-G-T genotype (Fig. 5A). In Genes & Health, the chr16-88716656-G-T$_T$ allele is associated with several traits related to red cell turnover (Supplementary Data 4), including bilirubin, ESR, haemoglobin, haematocrit, red blood cell mass, red cell distribution width, and serum potassium. This variant also showed weaker associations with random glucose ($P = 9.4 \times 10^{-6}$) and fasting glucose ($P = 3.0 \times 10^{-8}$) in the concordant direction (i.e. the T allele was associated with lower levels of both), however these associations did not surpass the study-wide significance threshold.

Previously reported associations of this SNP−derived from the subset of 8149 UKB participants of SAS ancestry−are also with red cell traits (haematocrit, haemoglobin and RBC) and with HbA1c[11,23]. The effect directions we report in the Genes & Health cohort are in the same effect direction, i.e. the chr16-88716656-G-T$_T$ allele was associated with increased haemoglobin and RBC, implying the rare (T) allele may protect against anaemia. Whole-genome sequencing data from the first 150,000 UKB genomes independently demonstrated the association of chr16-88716656-G-T$_T$ with haemoglobin in SAS ancestry individuals[24].

Although we excluded readings from individuals taking anti-hyperglycaemic medications, and so our results were largely obtained from controls or people who at the time of their blood tests were not diagnosed with diabetes, we sought to ensure that these findings were not confounded by disease status. To do so we conducted stratified GWAS of HbA1c in Genes & Health participants without diabetes ($N = 24,783$), excluding individuals diagnosed at any point during their electronic healthcare record follow-up. The strong association of chr16-88716656-G-T$_T$ with HbA1c persisted, and in fact strengthened in statistical significance in the non-diabetic cohort (Non-diabetics: $\beta = -0.64$, $P = 1.2 \times 10^{-181}$; beta orientated to the chr16-88716656-G-T$_T$ allele).

To quantify the association of chr16-88716656-G-T$_T$ with HbA1c and red cell traits in Genes & Health, we hard-called genotype dosages (the SNP is imputed with an imputation quality score [INFO] of 0.96), identifying 103 chr16-88716656-G-T$_{T/T}$ homozygotes (genotype frequency 0.2%), 3668 chr16-88716656-G-T$_{G/T}$ heterozygotes (7.2%) and 47,395 chr16-88716656-G-T$_{G/G}$ homozygotes (92.6%) in the entire cohort. Carriers of the T allele tended to have lower mean HbA1c values

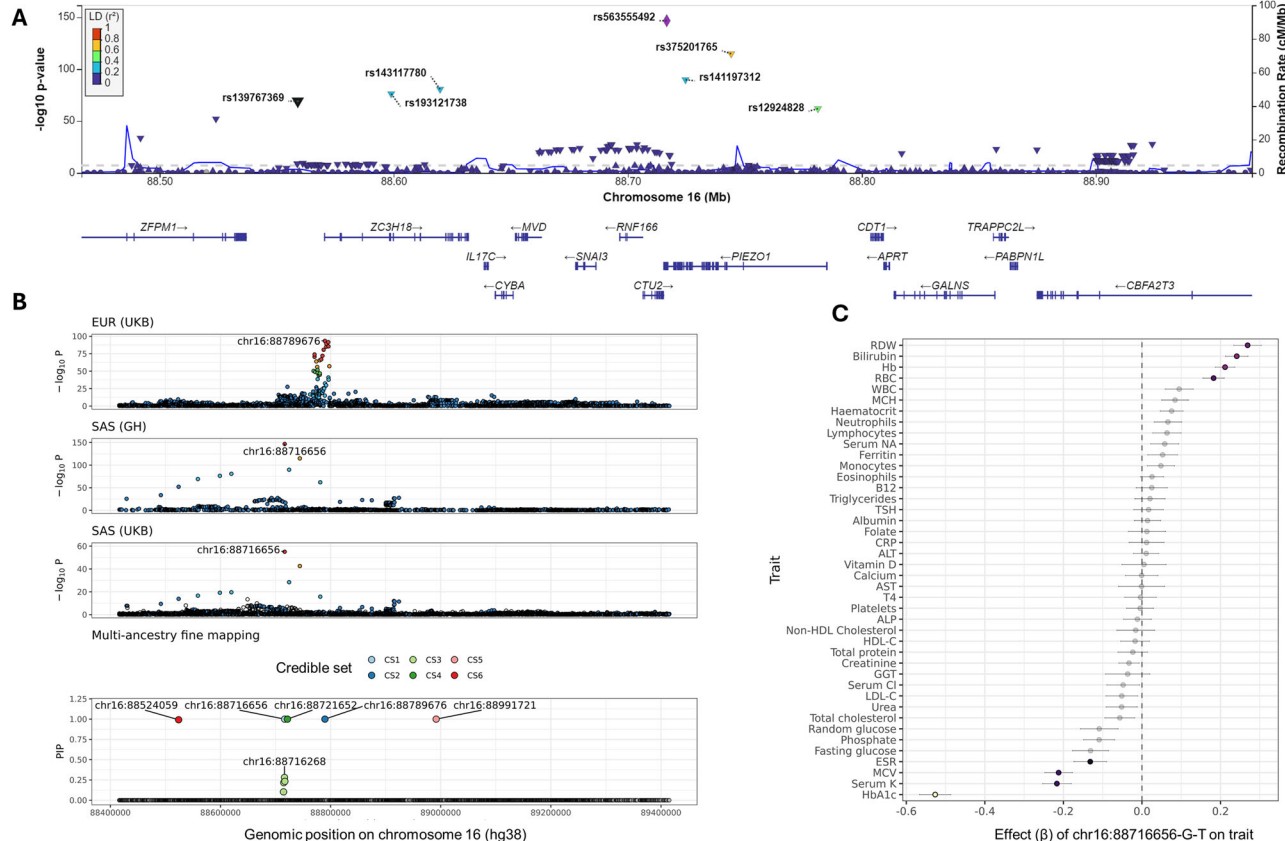

**Fig. 4 | Cross-ancestry meta-analysis and fine mapping at the *PIEZO1* locus for HbA1c. A** Locus plot showing the regional association of at the *PIEZO1* locus for HbA1c in the Genes & Health GWAS. The lead variant is coloured in purple, and other variants are coloured according to their strength of LD with the lead variant (derived from the 1000 Genomes reference samples of South Asian ancestry). **B** Panels display the regional association results at the *PIEZO1* locus on chromosome 16 from UKB EUR-ancestry GWAS (top, *N* = 400,825), Genes & Health SAS GWAS (middle, *N* = 30,967), UKB SAS-ancestry GWAS (third from top, *N* = 8329) and cross-ancestry fine mapping (SuSiEx, bottom). Points are coloured according to their LD with the lead variant in each ancestry for the top three panels, which is labelled with the SNP identifier. For the bottom panel, each of the independent credible sets is coloured separately and indicated in the legend. Of the six causal

signals, one was SAS-specific (indicated as 'CS1' on the plot), i.e. the causal signal had a posterior probability of >0.8 in the Genes & Health SAS and UKB SAS ancestries, but no other ancestry group in UKB (of EUR, AFR and EAS). This signal was mapped to a single causal variant, chr16-88716656-G-T. The top SNPs—i.e. the SNP with PIP > 0.1—are shown per credible set. Genomic co-ordinates are in hg38. **C** Forest plot showing pleiotropic association of the missense variant in *PIEZO1* (chr16-88716656-G-T; rs563555492) with various red cell traits in Genes & Health. Beta effect sizes represent the per-allele effect on rank-inverse normalised trait values in the Genes & Health GWAS. Error bars represent the 95% confidence interval, and are centred on the effect size estimate. Associations achieving study-wide significance are coloured in, the remainder are shown as translucent. Sample sizes differ by trait and are shown in Supplementary Data 3.

(N with HbA1c and genotype data, Mean HbA1c and SD per genotype: chr16-88716656-G-T$_{G/G}$ 40.8 mmol/mol [10.7], *N* = 28,666; chr16-88716656-G-T$_{G/T}$ 38.2 mmol/mol [11.1], *N* = 2270; chr16-88716656-G-T$_{T/T}$ 34.0 mmol/mol [6.8], *N* = 63). As the prevalence of T2DM was marginally lower in carriers of the T allele (25.5%, 23.0%, and 20.4% respectively for the G/G, G/T and T/T genotypes), and this variant had a suggestive effect on random glucose and fasting glucose in the Genes & Health GWAS, we considered whether the effect on HbA1c might be mediated via an impact on red cell survival, glycaemic control, or both. We defined the independent loci suggestively associated with both fasting and random glucose by LD-clumping SNPs with a suggestive impact ($P < 1 \times 10^{-5}$) and compared the impact of these SNPs on HbA1c. For both traits, the impact on HbA1c was disproportionately large (Fig. 5B, C), implying that while this SNP may act via both glycaemic and erythrocytic mechanisms, its effect is likely dominated by the latter.

To test this hypothesis, we identified a subset of the cohort with test results for both random glucose and HbA1c within 90 days of each other, and contrasted the average glucose within this time window with the measured HbA1c (*N* = 2956, of whom 228 carried at least one

chr16-88716656-G-T$_T$ allele). We found that in the first three quartiles of mean random glucose, HbA1c levels were significantly lower among carriers of the chr16-88716656-G-T$_T$ allele (Fig. 5D; unpaired *t* test *P* values: Q1 $1.3 \times 10^{-8}$, Q2 $4.9 \times 10^{-4}$, Q3 0.03, Q4 0.3), but there was no difference in those with high random glucose (>6.5 mM). We observed the same result (i.e. a significantly lower HbA1c for a given random glucose in quartiles 1–3, but not 4) when we restricted to individuals who never received a diagnosis of T2DM in their records (*N* = 1843), and when we repeated the analysis using fasting glucose readings (*N* = 5844). We confirmed this effect using linear models, modelling the effect on HbA1c of random or fasting glucose and genotype, adjusted for age at test, age², and gender. For both random and fasting glucose, we confirmed the expected effects of increasing age and male gender on higher HbA1c. Each 1 mM increase in fasting glucose was associated with a 5 mmol/mol increase in HbA1c, and presence of the T allele was associated with a 2.3 mmol/mol lower HbA1c ($P = 1.7 \times 10^{-11}$). For random glucose, each 1 mM increase was associated with a 2.7 mmol/mol increase in HbA1c, and presence of the T allele was associated with a 2.2 mmol/mol lower HbA1c ($P = 1.7 \times 10^{-5}$). We determined the expected increase in HbA1c for a given increase in fasting or random glucose

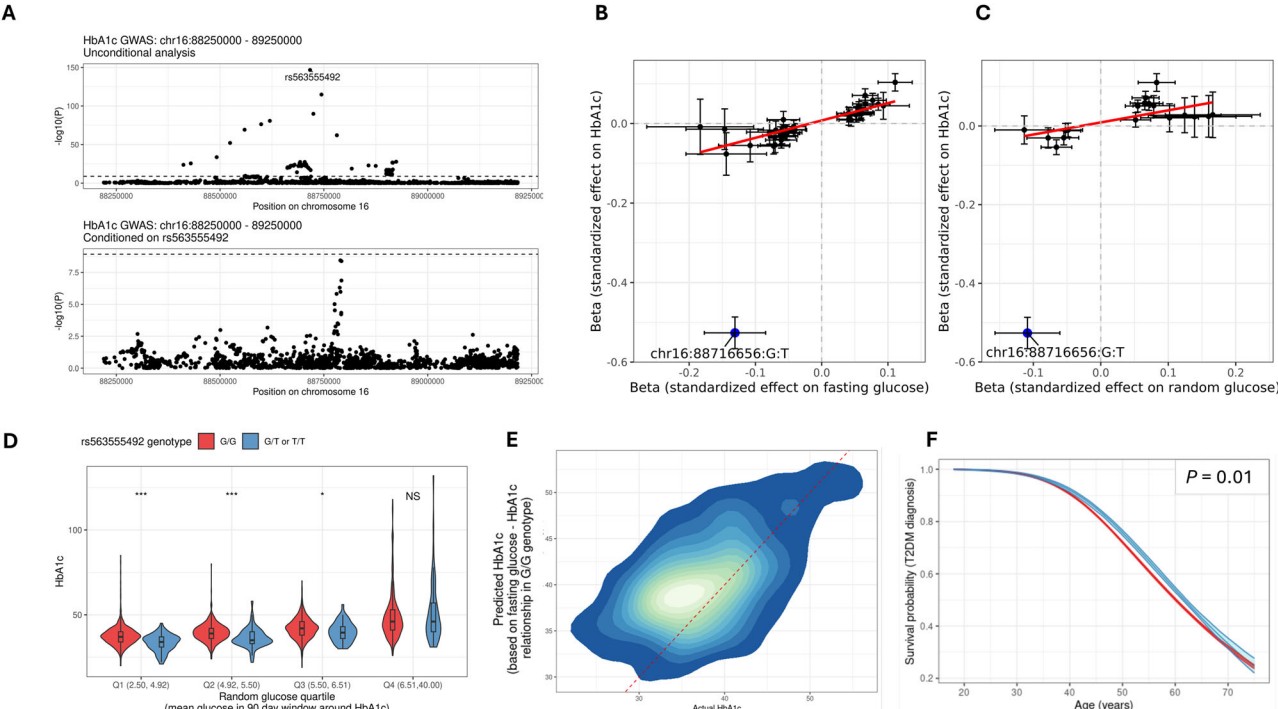

**Fig. 5 | The *PIEZO1* missense variant chr16-88716656-G-T influences HbA1c by affecting red blood cell lifespan, and is associated with a delay in T2DM diagnosis. A** Regional Manhattan plots showing association with HbA1c at the *PIEZO1* locus in the unconditional analysis (top panel) and after conditioning on chr16-88716656-G-T (bottom panel). No SNPs surpassed study-wide significance in the conditional analysis, confirming that chr16-88716656-G-T accounts for the majority of the signal at this locus. **B** Scatter plot showing the impact (i.e. GWAS effect estimate) of SNPs associated with fasting glucose (*N* for fasting glucose GWAS = 22,848) on HbA1c from the G&H GWAS. SNPs associated with fasting glucose at $P < 1 \times 10^{-5}$ are shown. The red line indicates a linear regression fit to the SNPs excluding the *PIEZO1* variant chr16-88716656-G-T, which is shown in blue. Error bars indicate the confidence intervals, centred on the effect size estimate. The scales refer to the GWAS effect scales, i.e. reflect rank-normalised traits. Although chr16-88716656-G-T was associated with lowered glucose, the reduction in HbA1c is greater than SNPs with an comparable impact on glucose. **C** As per **B**, but for random glucose rather than fasting glucose (*N* = 21,527). **D** Violin plots contrasting the HbA1c level within each quartile of random glucose for individuals with contemporaneous (i.e. within 90 days of each other) HbA1c and random glucose readings (*N* = 2956). *P* values are indicated as follows: \*\*\*<0.0005, \*\*<0.005, \*<0.05, NS ≥ 0.05. *P* values reflected unpaired *t* test comparisons between groups. Violins are coloured by genotype. The random glucose quartile is indicated alongside the range each quartile represents. The violin plot goes from the minimum to the maximum. The nested box plot shows the median, 25th and 75th centiles, and the whiskers extend from the minimum and maximum. **E** Density plot showing the observed vs expected HbA1c values among chr16-88716656-G-T_T carriers. The expected HbA1c value was derived from linear regression of fasting glucose on HbA1c among G/G homozygotes, adjusted for age and gender. The red dotted line indicates a smoothed 'expected' line. The density cloud shows that the majority of chr16-88716656-G-T_T carriers lie above this line, i.e. their expected HbA1c is greater than their actual HbA1c. **F** Survival curves showing the lower probability of T2DM diagnosis in carriers of the T allele, supporting the concept that this allele delays diagnosis.

using linear regression models (adjusted for the same covariates as above), and then applied this model to carriers of the T allele, allowing us to compare the observed vs expected HbA1c for each individual. Consistent with our hypothesis, most carriers of a T allele had a lower observed HbA1c than expected for their given level of glucose (chr16-88716656-G-T_G/T 266 / 399, 66.7%; chr16-88716656-G-T_T/T 10 / 11, 90.9%; figure 5E). We observed the same phenomenon for random glucose (i.e. lower than expected HbA1c in 70.1% and 100% of chr16-88716656-G-T_G/T and chr16-88716656-G-T_T/T groups respectively).

Given the HbA1c is widely used to diagnose diabetes, we reasoned that carriers of the chr16-88716656-G-T_T allele may tend to be diagnosed later due to disproportionately lowered HbA1c readings for any given level of average blood glucose. Of the 51,032 individuals included in this analysis (median 24.4 years of follow-up, total follow-up 1.3 million years), the median age at earliest T2DM diagnosis was 46 years old for carriers of the chr16-88716656-G-T_T allele (*N* = 862) and 45 years old for non-carriers (*N* = 12,040). We confirmed the impact of chr16-88716656-G-T_T on delaying T2DM diagnosis using a flexible parametric survival model[25], which showed a marginal estimate of median survival for chr16-88716656-G-T_T carriers 61.8 years vs 60.2 years for non-carriers, i.e. a delay, on average, of 1.6 years in those with the chr16-88716656-G-T_T allele (*P* = 0.01, Fig. 5F).

## Discussion

In this study of the Genes & Heath Cohort, we report the largest GWAS in a population of South Asian ancestry for a range of clinically relevant quantitative traits. We leverage linked EHRs and genotype data for ~50,000 British South Asian individuals of Pakistani and Bangladeshi ancestry to delineate the genetic architecture of these traits, explore the extent of cross-ancestry genetic overlap, improve fine-mapping resolution, and uncover how variants which are common in South Asian populations but rare in others may have a causal impact on complex traits. These discoveries underscore the power of genetic diversity in studies of complex traits and disease.

We demonstrate that while globally, there is significant overlap in the genetic architecture of these traits, there is heterogeneity of effect sizes at the level of individual variants. Using multi-ancestry fine mapping, we demonstrate that some of this heterogeneity is explained by causal variants which are common in South Asian populations and rare in others. We focus on the example of *PIEZO1*, in which a missense variant—chr16-88716656-G-T_T—is associated with lower HbA1c in persons of South Asian ancestry (-6 mmol/l difference between the two homozygote groups) but has no impact in other populations as it is ultra-rare. This variant appears to exert a disproportionately large effect on HbA1c for its effect on glycaemic control and is associated

with a lower HbA1c for any given level of glycaemic control. In keeping with this effect, we show that this variant is associated with a delayed onset of Type 2 Diabetes Mellitus. These results highlight the power of ancestrally-diverse datasets to interrogate the genetic basis of blood test parameters in health and disease.

The key strengths of our study include the large sample size of an ancestral population (British South Asian individuals) under-represented in genetic research[9] and the use of real-world electronic health data to provide robust phenotype definitions[26]. Studying complex trait genetics in this cohort of South Asian ancestry offers several unique advantages due to the high rates of common diseases such as T2DM[27], and the differences in allele frequency and LD compared with European-ancestry populations. The major limitation of our study is the lack of an external validation cohort of similar ancestral origin and potential bias due to non-random sampling, which is an inherent weakness of using routine healthcare data. We also use a single measure per individual—the mean—to summarise time-varying biomarkers as the outcome for GWAS. This approach has an attractive simplicity, and has the benefit of 'smoothing' outlying observations for the individual over time. The major weakness of this approach is that it discards the granularity of how these blood test parameters change over time. This is a question of major importance but is beyond the scope of the present study, and likely requires a bespoke approach for each trait to model longitudinal changes in these parameters using EHR data appropriately. Other limitations of our study regarding the phenotypes examined include the relatively small number of traits tested, the lack of clinical context for some tests (e.g. we cannot determine whether the observed lipid biomarkers were measured in a fasted state), and the static nature of these test results (e.g. we use random and fasting glucose, but do not have data on dynamic tests such as oral glucose tolerance tests). Validating the hypothesis that the *PIEZO1* missense variant chr16-88716656-G-T$_T$ has an appreciable impact on the validity of HbA1c will therefore require recall of participants and more detailed phenotyping. Further work is required to determine whether this marker might be clinically useful, for instance in designing genotype-stratified reference ranges for HbA1c.

Our findings represent the largest set of GWAS results from a population of South Asian ancestry for a range of traits, and emphasise the critical importance of diversity in studies of complex trait genetics.

## Methods
### Cohort
Genes & Health is a longitudinal genetic cohort study of over 50,000 individuals of South Asian ancestry living in the United Kingdom, with funding available to recruit up to 100,000 participants[28]. Individuals aged 16 and over from self-reported British Bangladeshi and British Pakistani backgrounds have been recruited since 2015. At recruitment, each participant completes a brief questionnaire, provides a saliva sample for genetic data, and consents to longitudinal linkage to primary and secondary care electronic health records alongside national databases such as NHS Digital, which contain linked Office for National Statistics mortality data. The demographic characteristics of this cohort have been previously described[28].

### Data sources
Values for quantitative trait phenotypes were extracted from routinely collected healthcare data. We obtained primary care data from the Discovery East London project, which encompasses seven clinical commissioning groups of primary care practices across inner and outer East London. These test results reflect tests requested by primary care clinicians. We also obtained test results from secondary care within the Barts Health laboratory dataset and Bradford Teaching Hospitals NHS Foundation Trust. These 'secondary care' datasets encompass tests processed in secondary care, but do not distinguish between tests performed as an inpatient, or as an outpatient in a clinic setting, and were ultimately excluded from the analysis (see below).

### Phenotype definitions and quality control
An overview of phenotypic quality control is shown in the supplement (Supplementary Fig. 1). Using these EHR data sources, we initially defined a broad range of quantitative traits from the primary care records of up to ~50,000 volunteers in the Genes & Health Study. We applied a stringent stepwise quality control procedure to derive individual-level phenotype data from the EHR. Briefly, after harmonising EHR data across the different data sources, we excluded test results which were non-numeric. Test results which contained 'greater than' or 'less than' indicators were simplified to represent this limit (e.g. a C-reactive protein of '<1' was relabelled as '1'). We then used custom codelists to define all occurrences of a test result in the EHR (codelists available in Supplementary data 2). For each trait, we defined a desired 'target unit' and converted all test results for the trait to have a common unit. Where test results had incompatible units—i.e. units which could not be converted with a simple multiplication factor— these results were excluded. We defined a manually curated minimum and maximum plausible value range for each trait influenced by prior clinical knowledge and reported ranges of these traits in UKB (where available). The full list of pre-specified minimum and maximum plausible values is given in the supplement (Supplementary Data 2). It is important to note that these ranges were devised to exclude values which are likely due to technical errors in sample processing or data entry, but to retain as much 'real' variation as possible. They are therefore somewhat deliberately broad. Trait values above or below the manually specified a priori limits of plausibility were excluded. Test results obtained from before the age of 18, after the date of data extract, or with a missing date were excluded. The age at test was approximated from the month and year of birth stated by the volunteer in their baseline questionnaire. In some cases, a single test result can appear in the EHR multiple times due to duplicate data entry mechanisms. We de-duplicated the data by excluding test results for the same volunteer with the exact identical value occurring within a 10-day rolling window.

To account for the impact of medications of quantitative traits, we first cleaned and curated prescription data from the primary care EHR. We then manually defined a set of commonly prescribed medications which can influence the traits studied in this paper, and defined the earliest prescription date per individual per drug. Rather than perform adjustment of values obtained while 'on-drug', we chose the pragmatic approach of restricting to data points obtained prior to drug initiation. While this led to a slight reduction in sample size for some traits and skewed the GWAS cohort towards a healthier population (by excluding readings in individuals on treatment with established disease), this approach led to stricter control of test statistic inflation in GWAS. The following drugs were accounted for: anti-hyperglycaemic agents (insulins, metformin, gliclazide, GLP-1 agonists, SGLT2 inhibitors) for glycaemic traits (HbA1c, random glucose), B12 (for vitamin B12), vitamin D (for serum vitamin D), folate (for serum folate), iron (for serum ferritin), statins (for cholesterol, LDL-C, HDL-C and triglycerides), thyroxine (for T4 and TSH). The list of traits adjusted for medications is shown in the supplement (Supplementary Data 2).

Blood test results for some assays in hospitalised inpatients are more likely to reflect transient abnormalities related to acute illness and so are less likely to be a reflection of steady-state biology. While the overall population mean values for primary and secondary care traits were highly correlated (Supplementary Fig. 2), there was substantially greater heterogeneity in the secondary care data (Supplementary Fig. 3). We therefore restricted the dataset to readings from primary care. Following exclusion of secondary care readings, we excluded outlying values more than 10-standard deviations from the

mean on the log-10 scale. This standard deviation threshold was chosen through an iterative process which aimed to maximise specificity (i.e. minimise the risk of including dubious test results reflecting errors in data entry, units, or failed assays) while retaining 'true' biological extremes. The resulting trait distributions were largely normally distributed on the log-10 scale, with a small number of exceptions (eosinophils, CRP and ESR, Supplementary Fig. 4). Intra-individual variation was low (Supplementary Fig. 5).

In addition to quantitative trait data, we defined the presence/absence of disease codes for diabetes for each participant by collating data from primary care, secondary care, and NHS Digital. We developed a custom pip package (https://github.com/genes-and-health/tre-tools) to streamline this process. We defined age at onset using the earliest recorded diagnostic code for each individual. Participants with no recorded codes for diabetes at any point during follow-up were considered unaffected controls.

### Residual-based adjustment of phenotypes

As both age at test and year of test explained a non-trivial amount of variation in several of the traits studied (Supplementary Fig. 6), we used a regression-based approach to account for these covariates upstream of genome-wide association testing. Using this approach allowed us to adjust each individual test result for these covariates prior to taking the mean per individual, and therefore provided additional control of confounding.

We used linear regression models adjusted for age at test, age at test$^2$, year of test, and year of test$^2$ to account for the effects of age at test and year of test. While the impact of age is likely to reflect both biological signal (i.e. age-related changes in various parameters) and confounding, the impact of year of test is more likely to be pure confounding due to technical noise, e.g. due to changes in laboratory assays over time. Importantly, in many cases the age at test differed substantially from the age at recruitment owing to the longitudinal nature of the healthcare record linkage. To ensure that model assumptions were satisfied, the outcome for the models was the log10-transformed trait value standardized by Z-scoring.

Applying this model to the post-quality control individual-level data yielded a residual trait value for each reading. Visual inspection of residual plots for each trait confirmed that this approach preserved the overall structure of the data. Where individuals had multiple readings for a trait, we took the mean of these residuals per individual as the outcome for GWAS. The mean residual per person was then further transformed using rank-inverse normalisation prior to association testing. Importantly, this procedure preserved the ranking of the distribution while destroying the underlying distribution on the original scale, and so the beta coefficients from GWAS cannot be straightforwardly interpreted as an absolute effect size.

### Trait selection

We defined a subset of 42 quantitative blood test traits with phenotypic data available for at least 10,000 participants with excellent data quality following application of stringent data quality control procedures. These traits covered a broad range of clinical indications, including full blood counts, haematinics, lipid metabolism, urea and electrolytes, liver function, markers of glycaemic control, markers of bone health, and thyroid function tests. The number of participants and individual readings varied substantially by test, however we chose to analyse traits with only data for >10,000 participants, i.e. focussing on the relatively 'routine' tests such as full blood counts and serum biochemistry. Genetic analyses of traits with low sample sizes are more prone to confounding by indication, i.e. bias caused by non-random sampling of the cohort, as may occur if a particular test is only requested in a specific disease context. For example, sex hormone tests may only be requested as part of investigation of subfertility, and so the distribution (and hence genetic architecture) may be distorted

compared to the true population genetic architecture. Our approach of focussing on relatively routinely acquired tests is expected to partially mitigate confounding due to non-random sampling. For the multi-ancestry analysis, we defined a subset of 29 traits from these 42 traits which overlapped with traits assessed in UKB and which we considered part of a 'routine' blood panel, i.e. not restricted to specific clinical indications.

### Genotyping

Genotyping was performed using genomic DNA extracted from saliva samples obtained via Oragene saliva sampling kits. Individuals were genotyped on the Illumina GSA v3 chip + extra multi-disease content. Genotype calling and initial genetic data quality control was carried out using Illumina GenomeStudio version 2.0. Briefly, automated clustering using the GenTrain algorithm was performed using 1970 selected very high-quality samples at a subset of high-quality variants (autosomal variants in Hardy–Weinberg equilibrium with GenTrain scores >0.7, reflecting high-confidence clustering). Iterative rounds of manual and automatic reclustering were then performed to identify low-quality variants and samples, ultimately resulting in a dataset with >99% call rate at 637,829 SNPs. This cluster file was then applied to the remaining ~50,000 samples to call genotypes. Samples were removed if they had lower call rates per sex than in the original batch (<99.2% for females, <99.5% for males). Of the 54,206 genotyped samples, individuals were removed due to a missing NHS number, discordant gender and genetic sex, implausible genetic duplicates (i.e. non-twin duplicates), resulting in a dataset of 51,176 individuals genotyped at 608,329 autosomal SNPs with a genotyping rate of >99.9%.

Following removal of rare variants (MAF < 0.0001), palindromic variants, and indels, genotypes were imputed to the TOPMED-r3 multi-ancestry imputation panel to genome build hg38 using the TOPMED imputation server[29]. Following imputation we performed SNP quality control, filtering to common (MAF > 0.01) biallelic autosomal variants with <10% missingness, imputation quality (INFO score) >0.7 with no significant ($P < 1 \times 10^{-15}$) deviation from Hardy–Weinberg equilibrium. Variants with duplicate positions were removed. Imputed dosages outside of the ranges 0–0.1, 0.9 –1.1 or 1.9 –2.0 were set to missing. We removed individuals with >10% missing genotypes. Genetic duplicate samples were identified with KING[30]–10 pairs of probable identical twins were identified, and one of each pair removed. Using principal component analysis, we identified and excluded <10 ancestral outliers who did not cluster with South Asian ancestry reference samples from the Human Genome Diversity Project and 1000 Genomes project[31]. Following a clustering-based procedure to estimate categorical ancestry groupings (Bangladeshi or Pakistani), a further 62 participants were excluded due to ambiguous ancestry. We further excluded participants with missing covariate (age and sex) information, and those not included in electronic healthcare record linkage.

### Genome-wide association testing

We conducted GWAS of mean biomarker values for the full set of 42 quantitative blood test traits using linear mixed models implemented in REGENIE[32]. For step 1 of REGENIE we used the full set of common (MAF > 0.01) genotyped markers. We adjusted for the following covariates in GWAS: age at recruitment, age at recruitment$^2$, sex, age at recruitment × sex, inferred genetic ancestry (Bangladeshi or Pakistani ancestry), and the first 20 genetic principal components. Association testing (step 2) was performed using default parameters, adjusting for the same covariates as step 1. GWAS of HbA1c in non-diabetics was performed in exactly the same way as in the primary analysis except for the exclusion of individuals with any diabetes codes in their EHR records. The outcome of the GWAS in each case was the mean of the age-adjusted and year-adjusted blood test value per individual, transformed via rank-inverse normal transformation (see above for details of phenotype pre-processing).

## UK Biobank GWAS

To compare and combine our GWAS results with those from UKB, we obtained GWAS summary statistics from the pan-UKB multi-ancestry project (https://pan.ukbb.broadinstitute.org). Based on NHS number, approximately 0.05% of Genes & Health participants have also taken part in UK Biobank, and so participant overlap is unlikely to introduce significant bias. Of the 42 blood-based tests in our initial GWAS, we selected a subset of 29 traits which overlapped with UK Biobank (Supplementary Data 6). The pan-UKB project inferred the genetic ancestry of UK Biobank participants and categorised participants into one of six continental superpopulations: African ($N = 6636$), Admixed ($N = 980$), South Asian ($N = 8876$), East Asian ($N = 2709$), European ($N = 420,531$), and Middle-Eastern ($N = 1599$). Within each ancestry group, pan-UKB conducted GWAS of a range of phenotypes, including 29 blood tests which overlapped with our dataset. These blood test values were derived from baseline samples obtained on study entry[33], and therefore represent systematically collected one-off data points, in contrast to the longitudinal routine EHR data used for GWAS in our study. All phenotypes were transformed using rank-inverse normalisation within each ancestral group prior to GWAS.

## Variant annotation and identification of cohort-specific loci

To identify SNP-trait associations which were specific to the Genes & Health cohort, we first defined independent loci by LD-clumping the GWAS results using samples of South Asian ancestry ($N = 489$) from the 1000 Genomes reference panel. For each SNP associated with any trait in the Genes & Health cohort at a study-wide significance threshold ($P = 5 \times 10^{-8}/42$ traits $= 1.2 \times 10^{-9}$) we identified all SNPs in LD ($R^2$ of $>0.001$) within a 1-MB window and considered this a single locus. The SNP with the lowest $P$ value in each LD window was labelled the lead SNP. SNPs were annotated with the nearest gene using the Variant Effect Predictor (VEP) tool[34]. To explore whether any of these loci were cohort-specific, we defined a broad window (± 1MB) around the lead variant in the Genes & Health GWAS for each SNP-trait association. We then searched within this window for any suggestive signals in the pan-UKB multi-ancestry GWAS ($P_{UKB} < 1 \times 10^{-5}$) in the major ancestry groups (EUR− European, SAS−South Asian, AFR−African, EAS−East Asian)[35,36]. If there were no signals in any UKB population within this window at $P_{UKB} < 1 \times 10^{-5}$, we considered the locus to be GH-specific. This definition was deliberately conservative to minimise the risk of type 1 error. To identify variants which were detected in GH but rare in other ancestries, we used VEP to look up reference allele frequencies (from the 1000 Genomes Project & gnomAD) of the variants achieving statistical significance in GH.

## Test statistic inflation & genetic correlation between traits and ancestries

Test statistic inflation was estimated initially using $\lambda_{GC}$ (median $\chi^2$ statistic/median $\chi^2$ of a 1-degree of freedom distribution), and subsequently with LDSC[37] to disentangle inflation due to polygenicity versus population stratification. In the absence of an established LD score reference panel for South Asian ancestry individuals, LD scores were estimated within the Genes & Health cohort. Cross-trait genetic correlation was calculated as LDSC-estimated genetic correlation ($r_g$) for all trait pairs. We used Popcorn[38] to estimate cross-ancestry genetic correlation (the genetic effect) accounting for differences in allele frequency and LD structure for the 29 traits overlapping between Genes & Health and UK Biobank. For the cross-ancestry correlations, LD scores were obtained from the 1000 Genomes reference samples of South Asian ($N = 489$) and European ($N = 503$) ancestry respectively. We used common SNPs (MAF $> 0.05$) for this analysis.

## Multi-ancestry meta-analysis

We meta-analysed GWAS summary statistics for the 29 traits overlapping between our study and the pan-UKB GWAS. Cross-ancestry meta-analysis was performed using meta-regression implemented in MR-MEGA[24]. We ran MR-MEGA using seven separate ancestral populations: the Genes & Health cohort of SAS ancestry and the six ancestral groups defined in pan-UKB (EUR, Central/South Asian [CSA], EAS, Middle Eastern [MID], AFR, and admixed American [AMR]). Although similar population groups (e.g. the SAS cohort) are likely to be correlated in terms of genetic ancestry between studies, we chose to treat these groups separately due to the expected sub-continental differences in genetic ancestry between cohorts. UKB summary statistics were lifted to hg38 using LiftOver[39]. We used four axes of genetic variation to model the differences in genetic ancestry between GWAS cohorts as these appeared to distinguish the known ancestral composition of the cohorts and the maximum permitted by MR-MEGA is four (number of cohorts−3). Genomic control statistics (lambda) were calculated separately for each cohort and adjusted for in the meta-analysis. We considered variants tested in all seven cohorts, with an average (weighted) effect allele frequency of between 1 and 99% across all cohorts, and with no evidence of 'residual' heterogeneity, i.e. heterogeneity in effect size between studies not explained by the ancestral axes of variation ($P_{Heterogeneity-Residual} < 0.05$). We identified loci with evidence of differences in allelic effects between ancestry using the $P$ value for heterogeneity due to ancestry. We prioritised loci for fine mapping if they achieved statistical significance in the GWAS meta-analysis at $P < 1.72 \times 10^{-9}$ and demonstrated evidence of ancestral heterogeneity at $P < 1.72 \times 10^{-9}$.

## Fine mapping

We performed multi-ancestry fine mapping using SuSiEx[40]. Reference sample data from 1000 Genomes was downloaded from the MAGMA website (https://ctg.cncr.nl/software/magma). We aggregated GWAS summary statistics in hg38 build from the pan-UKB resource (for each of the major ancestral groups: EUR, AFR, SAS, EAS) and from G&H (SAS). We fine mapped within a 1MB window, using the 500KB either side of the lead SNP (i.e. the SNP with the lowest $P$ value for association in the cross-ancestry meta-analysis). We included SNPs with MAF ≥ 0.01 in at least two of the included GWAS studies (i.e. Genes & Health SAS-ancestry and the EAS, AFR, EUR, and SAS-ancestry GWAS from UKB). Each individual GWAS was filtered to common (MAF ≥ 0.01) biallelic SNPs prior to fine mapping. We considered only those credible sets with at least one SNP at $P < 1.72 \times 10^{-9}$ in at least one cohort. SuSiEx estimates the probability that a fine-mapped signal is causal in a given population[40]: we considered signals to be causal in a population if the post-hoc probability for the credible set in that population exceeded 0.8. We defined SAS-specific causal variants as those credible sets with a post-hoc probability of >0.8 in both the SAS cohorts (GH and UKB), but <0.8 in the other tested cohorts (UKB EUR, AFR and EAS). The locus plot of the PIEZO1 locus association with HbA1c in the Genes & Health cohort was generated with LocusZoom using 1000 Genomes reference data from individuals of South Asian ancestry[41].

## Conditional analysis

To determine whether the SAS-specific signals identified in multi-ancestry fine mapping at the PIEZO1 and HBB loci represented single causal variants which were excluded due to very low frequency in non-SAS cohorts, we first performed stepwise forward conditional analysis at these loci using a broad positional definition of ±1MB either side of the lead SNP using conditional and joint analysis implemented in GCTA[42,43]. For this analysis we used 'in-sample' LD, i.e. LD estimated from the Genes & Health samples themselves, and used a study-wide significance threshold of $P < 1.72 \times 10^{-9}$ to define independent SNPs. We validated these results by performing single-ancestry fine mapping under the single causal variant assumption. Fine mapping was performed with a positional definition of the loci (±1MB either side of the lead SNP) using FINEMAP[44]. We report the posterior inclusion probability calculated for this configuration (i.e. single causal variant).

## Impact of PIEZO1 variant on diabetes age at onset

To explore the impact of chr16-88716656-G-T$_T$ on the age at T2DM diagnosis, we defined a longitudinal cohort using linked primary and secondary care data for all genotyped GH participants ($N = 51,1176$). Follow-up was defined as starting at age 18 and end of follow-up was defined as the earliest of the following: death (ascertained via linked ONS registered all-cause mortality), end of study (July 2023), or the earliest record of a T2DM diagnosis (defined using the same custom codelist as used for the T2DM-stratified GWAS). People with T2D diagnosed before age 18 ($N = 73$), or ambiguous ancestry ($n = 62$) were removed. We initially used Cox proportional hazards to model the effect of chr16-88716656-G-T genotype (encoded in a dominant fashion due to the low number of rare homozygotes) on time to T2DM diagnosis, adjusting the model for gender, inferred genetic ancestry (Pakistani or Bangladeshi), and the first 20 genetic principal components. We used age (years) as the timescale with study entry (truncated) at age 18. We tested the proportional hazards assumption using the Schoenfeld test, observing that the proportional hazards assumption did not hold (i.e. Schoenfeld $P < 0.05$) for chr16-88716656-G-T. Therefore, we used a flexible parametric survival model (restricted cubic splines) with a time-dependent effect for genotype, selecting the appropriate number of knots via model fit indices (Akaike Information Criterion) and visual inspection of modelled vs observed survival curves. Confidence intervals were obtained through bootstrapping. The predicted median survival (dependent on genotype, standardized for all other model parameters), was obtained directly from the fitted model.

### Reporting guidelines

This study was conducted in accordance with the STREGA guidelines for genetic association studies.

### Ethical approval

This research was conducted under an approved application to use the Genes & Health resource Genes & Health was approved by the London South East NRES Committee of the Health Research Authority (14/LO/1240).

### Computing

Analyses were conducted using R version 4.2.3 in the Genes & Health Google Cloud Trusted Research Environment (TRE) and R version 4.2.2 on the Queen Mary High Performance Computing cluster[45].

### Reporting summary

Further information on research design is available in the Nature Portfolio Reporting Summary linked to this article.

## Data availability

Individual-level data from Genes & Health are available for bona fide researchers on application (https://www.genesandhealth.org/). Genome-wide association study summary statistics for all quantitative traits tested are publicly-available for download with this paper using the instructions here (https://github.com/benjacobs123456/genes_and_health_quant_gwas?tab=readme-ov-file#download-gwas-summary-stats), can be browsed using the interactive Shiny app here: https://benjacobs.shinyapps.io/gh_quant_trait_gwas_browser/, or can be downloaded via the GWAS catalogue https://www.ebi.ac.uk/gwas/ (GCP ID GCP000997, study accessions: GCST90448597, GCST90448598, GCST90448599, GCST90448600, GCST90448601, GCST90448602, GCST90448603, GCST90448604, GCST90448605, GCST90448606, GCST90448607, GCST90448608, GCST90448609, GCST90448610, GCST90448611, GCST90448612, GCST90448613, GCST90448614, GCST90448615, GCST90448616, GCST90448617, GCST90448618, GCST90448619, GCST90448620, GCST90448621, GCST90448622, GCST90448623, GCST90448624, GCST90448625, GCST90448626, GCST90448627, GCST90448628, GCST90448629, GCST90448630, GCST90448631, GCST90448632, GCST90448633, GCST90448634, GCST90448635, GCST90448636, GCST90448637, and GCST90448638).

## Code availability

All code used for the analysis is available at github.com/benjacobs123456/genes_and_health_quant_gwas or via Zenodo (https://doi.org/10.5281/zenodo.13221412).

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

## Acknowledgements

Genes & Health is/has recently been core-funded by Wellcome (WT102627, WT210561), the Medical Research Council (UK) (M009017, MR/X009777/1, MR/X009920/1), Higher Education Funding Council for England Catalyst, Barts Charity (845/1796), Health Data Research UK (for London substantive site), and research delivery support from the NHS National Institute for Health Research Clinical Research Network (North Thames). Genes & Health is/has recently been funded by Alnylam Pharmaceuticals, Genomics PLC; and a Life Sciences Industry Consortium of Astra Zeneca PLC, Bristol-Myers Squibb Company, GlaxoSmithKline Research and Development Limited, Maze Therapeutics Inc, Merck Sharp & Dohme LLC, Novo Nordisk A/S, Pfizer Inc, Takeda Development Centre Americas Inc. We thank Social Action for Health, Centre of The Cell, members of our Community Advisory Group, and staff who have recruited and collected data from volunteers. We thank the NIHR National Biosample Centre (UK Biocentre), the Social Genetic & Developmental Psychiatry Centre (King's College London), Wellcome Sanger Institute, and Broad Institute for sample processing, genotyping, sequencing and variant annotation. We thank: Barts Health NHS Trust, NHS Clinical Commissioning Groups (City and Hackney, Waltham Forest, Tower Hamlets, Newham, Redbridge, Havering, Barking and Dagenham), East London NHS Foundation Trust, Bradford Teaching Hospitals NHS Foundation Trust, Public Health England (especially David Wyllie), Discovery Data Service/Endeavour Health Charitable Trust (especially David Stables), Voror Health Technologies Ltd (especially Sophie Don), NHS England (for what was NHS Digital)—for GDPR-compliant data sharing backed by individual written informed consent. We would also like to acknowledge Hye In Kim, Jonathan Davitte, and Karol Estrada who contributed to discussions regarding strategies for phenotype data quality control. B.M.J. was primarily supported by a Medical Research Council (MRC) Clinical Research Training Fellowship (CRTF) jointly funded by the UK Multiple Sclerosis Society (MR/V028766/1). B.M.J. has also received support from the National Multiple Sclerosis Society, Barts Charity, and AIMS2CURE. DS is funded by the Tackling Multimorbidity at Scale Strategic Priorities Fund programme (MR/W014416/1) delivered by the Medical Research Council and the National Institute for Health Research in partnership with the Economic and Social Research Council and in collaboration with the Engineering and Physical Sciences Research Council. S.H. is supported by a Wellcome Trust HARP Fellowship (227532/Z/23/Z).

## Author contributions

B.M.J., D.S., S.H., and D.V.H. conceived of the study. B.M.J., D.S., S.H., Miriam Samuel and J.Z. curated the quantitative traits used as phenotype outcomes, with supervisory help from DVH and RM. BMJ and SH ran the genetic analysis. All authors—B.M.J., D.S., S.H., J.Z., M.S., S.K., S.B., K.W., C.L., R.D., S.F., C.M., M.K.S., H.M., M.P., R.M., and D.V.H.—reviewed the manuscript for important intellectual content and take ownership of its findings. DVH supervised this work with input from M.P., Moneeza Siddiqui, H.M., and R.M.

## Competing interests

The authors declare no competing interests.

## Additional information

## Genes & Health Research Team

Shaheen Akhtar[9], Mohammad Anwar[10], Elena Arciero[9], Omar Asgar[11], Samina Ashraf[12], Saeed Bidi[13], Gerome Breen[14], James Broster[13], Raymond Chung[14], David Collier[13], Charles J. Curtis[14], Shabana Chaudhary[13], Megan Clinch[13], Grainne Colligan[10], Panos Deloukas[13], Ceri Durham[10], Faiza Durrani[13], Fabiola Eto[13], Sarah Finer[13], Joseph Gafton[13], Ana Angel Garcia[13], Chris Griffiths[13], Joanne Harvey[13], Teng Heng[9], Sam Hodgson[13], Qin Qin Huang[9], Matt Hurles[9], Karen A. Hunt[13], Shapna Hussain[13], Kamrul Islam[13], Vivek Iyer[9], Ben Jacobs[13], Ahsan Khan[13], Cath Lavery[13], Sang Hyuck Lee[14], Robin Lerner[13], Daniel MacArthur[15], Daniel Malawsky[9], Hilary Martin[9], Dan Mason[16], Rohini Mathur[13], Mohammed Bodrul Mazid[13], John McDermott[17], Caroline Morton[13], Bill Newman[17], Elizabeth Owor[13], Asma Qureshi[13], Samiha Rahman[13], Shwetha Ramachandrappa[14], Mehru Reza[13], Jessry Russell[13], Nishat Safa[13], Miriam Samuel[13], Michael Simpson[14], John Solly[13], Marie Spreckley[13], Daniel Stow[13], Michael Taylor[13], Richard C. Trembath[14], Karen Tricker[13], Nasir Uddin[13], David A. van Heel[13], Klaudia Walter[9], Caroline Winckley[18], Suzanne Wood[13], John Wright[19] & Julia Zollner[3]

[9]Wellcome Sanger Institute, London, UK. [10]Social Action for Health, London, UK. [11]Manchester University NHS Trust, Manchester, UK. [12]Bradford Teaching Hospitals, Bradford, UK. [13]Queen Mary University of London, London, UK. [14]King's College London, London, UK. [15]Garvan Institute of Medical Research, Darlinghurst, NSW, Australia. [16]Born in Bradford, Bradford, UK. [17]University of Manchester, Manchester, UK. [18]NIHR Clinical Research Clinical Trials, Manchester, UK. [19]Bradford Institute for Health Research, Bradford, UK.

