## [Peer Review File · Nature Communications]

Genetic architecture of routinely acquired blood tests in a British South Asian cohortREVIEWER COMMENTS

Reviewer #1 (Remarks to the Author):

Jacobs*, Stow*, Hodgson*, Zöllner*, Samuel* et al. reports genome-wide association mapping of clinically relevant blood biochemistry traits in South Asian ancestry groups by analyzing up to 37,352 individuals in the Genes & Health cohort (G&H) in the UK, which collects the genotype data with linked electronic health record information. The manuscript is overall well written, but several sections would benefit from further clarification. The strength of their study is that they report several genome-wide association signals that were not previously reported in the literature. I have several questions on the methodologies used in the analysis and comments to further improve the clarity of the presentation of their results. Overall, I believe a minor revision of the manuscript would be sufficient for the publication of their study.

1. The authors have already shown great efforts to demonstrate the quality of their phenotypic measurements. Given the non-random sampling of the phenotypes, I believe there is still some additional analysis and quality control needs to be done. What is the distribution of the number of blood biochemistry measurements across individuals? If some individuals have an extremely large number of measurements, it may perhaps reflect the underlying disease status that necessitated the repeated measurements of the trait and perhaps such extreme outlier individuals should be excluded from the association analysis. Similarly, did the author assess/adjust for the length of the record length (number of the event and years of the record) when conducting association analysis? The authors showed that the measurements from each individual across different time points are highly correlated ($p < 10^{-10}$, indicating the non-zero correlation of the measurements), but do not show whether the intra-individual correlation is higher than inter-individual correlations (perhaps permutation test p-value would be appropriate).
2. The discovery of the genome-wide association signals that was not previously reported in the literature is one of the strengths of the study. The authors did a great job in evaluating whether the association signals were replicated in other cohorts, such as UK Biobank. However, I am puzzled to see the rs7314285(CUX2)-VitD association was in the African and European ancestry groups in UK Biobank (UKB), but not in South Asian and East Asian ancestry groups in UK Biobank. Is it simply because the simplistic description of “South Asian” in UKB and G&H does not reflect the heterogeneity in the ancestry groups? Or, is it stemming from the fact that G&H collects blood biochemistry from clinical settings whereas UKB measured biochemistry from blood samples collected at the assessment center? Please provide more description on why or why there is no replication of the findings on the rs7314285(CUX2)-VitD association in UKB’s “South Asian” cohort.
3. Related to the point above, are the two sets of summary statistics from UKB and G&H directly comparable in terms of the phenotypic transformation? Given the skew of the distributions, it is common to apply log-transformation of the phenotype before conducting the GWAS analysis. I wonder if the lack of the replication of some of their findings can also be explained by the difference in phenotypic transformation.

4. At the CUX2 locus, authors report the moderate linkage of two SNPs (rs7314285 and rs73413596) in South Asian. What are the LD statistics in European individuals? Also, did the authors confirm if the “novel” rs7314285(CUX2)-VitD association is significant after applying conditional analysis on rs73413596?

5. In the description of the rs7314285(CUX2)-VitD association, the authors mention that the variant is also associated with sex hormone binding globulin and testosterone levels in UK Biobank. Are those fine-mapped causal variants? What is the relevance of the pleiotropic associations in interpreting the reported “novel” associations? Please clarify in the main text.

6. The use of the term “novel” may not be appropriate. Associations previously not reported in the literature (or in UK Biobank) would be more appropriate.

7. Regarding the methodology for the fine-mapping analysis, what is the “post-hoc probability for the credible set”? Please clarify in the text. Also, the multi-ancestry fine mapping was performed on SuSiEx, but the single-ancestry fine mapping was conducted with FINEMAP, not SuSiE. Please describe what criteria did the authors consider when selecting the types of fine mapping software.

8. For the PEIZO1-ZFPM1 locus analysis, the posterior inclusion probability of the SAS-specific causal variants should be described in the main text.

9. Regarding the SAS-specific variants, rs563555492, did the author know why the variant is monomorphic in AFR, EUR, and EAS populations? Is there any selection happening at the locus in SAS?

10. The abstract would benefit from improving the clarity. For example, the authors claim their results “illustrates how ancestry-specific genetic variants may be of relevance for the design of clinical reference ranges”. However, there are not substantial results on the design of clinical reference ranges in the main text. The authors also report discovery of genome-wide association signals specific to Gene & Health, but they do not mention that “Gene & Health” is a name of the cohort. The description on the South Asian specific associations on rs563555492 may benefit from including effect size and posterior inclusion probability.

11. Figures and Table would benefit from further clarification. For example, the error bars in Figure 1B are not defined. The numbers in parentheses for “age at recruitment” in Table 1 is not defined. The abbreviations, for example, EAF (effect allele frequency, in Fig 1B) and PIP (posterior inclusion probability in Fig 2A) are not explicitly defined. In the “variance explained by covariates” plot, the columns are shown in alphabetical order, but the “combined” should be shown at the left-most column or the right-most column.

12. The sample size of the association study presented in the manuscript is not clearly presented in the main text. In the abstract, the authors claim the “largest” genome-wide association study, but sample size is not provided in the abstract itself. In the introduction, the authors say “Here we report ... a longitudinal genotype-phenotype study comprising ~50,000 individuals ...,” but Table 1

legend says the individuals with both genetic and phenotypic data are up to $n=37,352$ individuals. The authors should clarify the sample size analyzed in the study across the multiple relevant sections of the manuscript, given that the GWAS discovery from one of the largest South Asian cohorts is the central takeaway message of the manuscript.

13. Minor points. The phenotype curation process has duplicated descriptions. For example, the selection of 39 traits based on having at least 50% of high-quality measurements are described twice. Under the “genotyping” subsection in the methods section, biological “sex” should be used instead of “gender”. Under the “GWAS testing” subsection in the methods section, “IQR 15%” does not represent a “range.” The sample size of the LD reference panel used in LD-clumping is not described. Under the “genetic correlation” subsection, the authors say “inter-trait heritability” but it should have been written as “genetic correlation.” Also, the reference for Popcorn used for trans-ancestry genetic correlation is missing.

Reviewer #2 (Remarks to the Author):

The paper uses data from the Genes & Health study for a genome-wide association study of 39 blood test traits obtained from linkage to Electronic Healthcare Records (EHR) of about 40,000 British Bangladeshi and British Pakistani individuals. Main findings are eight new significant associations in Genes & Health but not identified in the UK Biobank. The manuscript has major issues as detailed below. It is not clear which loci are novel and replication were not attempted in populations of South Asians.

Some major concerns:

1. The paper description is somewhat unusual as it includes results (Table 1) in the methods section, and discussion in results.
2. It looks like the study included blood tests that cannot be distinguished as inpatient or outpatient. This distinction is important as inpatient lab tests relate to transitory shifts due to acute disease and are very different from chronic laboratory values obtained in outpatient settings. This is important because they used mean values across multiple dates.
3. The description of the procedures for quality control of the data needs to be clearly described for each laboratory data given their EHR source. It will be important to know fasting status (lipids, glucose), and if participants were taking medications (diabetics taking insulin, and so on). How this compares to procedures used for lab data in published studies (transformation, covariates). For example, HbA1c and glucose should be restricted to individuals without diabetes.
4. A better context on why we should study the genetics of these traits in South Asians needs to be included related to burden of disease or known population genetic findings. Provide some background on the admixture of South Asians in Britain.
5. It is not clear which of these loci are novel as the comparisons seem to be made with the UK Biobank and not large published GWAS. Replication in South Asians in the UKB would be the preferred step.

Other comments

Explain why you used 20 PCs given little observed inflation.

A sensitivity analysis using just primary care outpatient labs is needed.

Heritability seems too high for some traits, so include the range for heritability in the literature in supplemental table for comparison.

Not sure why the investigators do not use their own data LD for clumping. The overlap with UK Biobank findings should be defined using EUR LD instead of bp range.

For MR-MEGA meta regression, it is important to know if the UK Biobank and your study used the same phenotype transformations and statistical methods because you are combining the summary data.

What is the overlap between the UK Biobank SAS participants and those from the Genes & Health study?

Clarify if the new findings in the Genes & Health study were replicated in the UK Biobank South Asians. It seems that the comparisons were only made for EUR and AFR UKB participants.

For the Vit D CUX2 locus, conditional analysis is needed for the published variant in the region.

For the PIEZO1 locus, the heterogeneity could be due to trait selection (lack of exclusion of diabetics).

Supplemental figures do not have legends, neither the supplemental tables.

Response to reviewers

Reviewer #1 (Remarks to the Author):

1. The authors have already shown great efforts to demonstrate the quality of their phenotypic measurements. Given the non-random sampling of the phenotypes, I believe there is still some additional analysis and quality control needs to be done. What is the distribution of the number of blood biochemistry measurements across individuals? If some individuals have an extremely large number of measurements, it may perhaps reflect the underlying disease status that necessitated the repeated measurements of the trait and perhaps such extreme outlier individuals should be excluded from the association analysis. Similarly, did the author assess/adjust for the length of the record length (number of the event and years of the record) when conducting association analysis? The authors showed that the measurements from each individual across different time points are highly correlated ($p < 10^{-10}$, indicating the non-zero correlation of the measurements), but do not show whether the intra-individual correlation is higher than inter-individual correlations (perhaps permutation test p-value would be appropriate).

We have now included a detailed supplementary note which describes the phenotype data. As the reviewer suggests, we do indeed see relationships between the number of readings and the average reading per individual. This is likely to reflect non-random sampling, i.e. people with a disease or with an abnormal blood test result are more likely to have it repeated. We also see a relationship between mean value and the duration of the individual's record (i.e. the time from their earliest to latest reading) for most traits.

We disagree with excluding these individuals as they are likely to represent 'biological' outliers i.e. people with extreme phenotypes who add power to the GWAS as they reflect genuine disease entity / extreme phenotype states. For instance, the patients with coded Type 2 Diabetes in the cohort have more HbA1c readings than non-diabetics: adjusting for the frequency of recording or excluding these people would dampen down real signal coming from the diabetic cohort. Rather than excluding these cases, we show the impact of these confounders so that the reader can clearly see the heterogeneity in these data and the possible confounding by nonrandom sampling.

2. The discovery of the genome-wide association signals that was not previously reported in the literature is one of the strengths of the study. The authors did a great job in evaluating whether the association signals were replicated in other cohorts, such as UK Biobank. However, I am puzzled to see the rs7314285(CUX2)-VitD association was in the African and European ancestry groups in UK Biobank (UKB), but not in South Asian and East Asian ancestry groups in UK Biobank. Is it simply because the simplistic description of "South Asian" in UKB and G&H does not reflect the heterogeneity in the ancestry groups? Or, is it stemming from the fact that G&H collects blood biochemistry from clinical settings whereas UKB measured biochemistry from blood samples collected at the assessment center? Please provide more description on

why or why there is no replication of the findings on the rs7314285(CUX2)-VitD association in UKB's "South Asian" cohort.

With the addition of new samples, restriction to primary care data only, and slight change in covariates (control for Bangladeshi vs Pakistani ancestry), this signal has dissipated to below genome-wide significance in the Genes and Health cohort and thus has been removed from the manuscript. We expect the signal we saw in the smaller cohort was a type 1 error. We have implemented a study-wide significance threshold adjusting for the number of phenotypes which provides additional security against further false positives.

In our revised analysis, this variant had a beta of 0.06 and a P value of 1×10^{-5} . We think that this change is due to a reduction in measurement error by restricting to primary care samples.

3. Related to the point above, are the two sets of summary statistics from UKB and G&H directly comparable in terms of the phenotypic transformation? Given the skew of the distributions, it is common to apply log-transformation of the phenotype before conducting the GWAS analysis. I wonder if the lack of the replication of some of their findings can also be explained by the difference in phenotypic transformation.

While differences in methods of sample processing, clinical context, genetic batch effects, and ancestry may explain the discrepancy, we believe that statistical transformation was unlikely to contribute to these differences. In our study and the pan-UKB GWAS, the outcome was rank-inverse normalised prior to GWAS, coercing the distribution to a normal distribution which preserves the ranks of the original observations. Thus log transformation prior to rank normalisation would not lead to any difference.

Here is an empirical demonstration in our data - taking the log10 of the neutrophil count prior to applying the rank-inverse normalisation does not affect the outcome:

We have clarified in the methods that we use this transformation prior to GWAS.

4. At the *CUX2* locus, authors report the moderate linkage of two SNPs (*rs7314285* and *rs73413596*) in South Asian. What are the LD statistics in European individuals? Also, did the authors confirm if the “novel” *rs7314285*(*CUX2*)-VitD association is significant after applying conditional analysis on *rs73413596*?

Please refer to the response to reviewer 1 point 2. Since the locus did not persist in the larger sample, we have therefore removed this finding from the paper.

5. In the description of the *rs7314285*(*CUX2*)-VitD association, the authors mention that the variant is also associated with sex hormone binding globulin and testosterone levels in UK Biobank. Are those fine-mapped causal variants? What is the relevance of the pleiotropic associations in interpreting the reported “novel” associations? Please clarify in the main text.

Please refer to the response to reviewer 1 point 2.

6. The use of the term “novel” may not be appropriate. Associations previously not reported in the literature (or in UK Biobank) would be more appropriate.

We have amended the word novel to ‘cohort-specific’ or ‘specific to Genes and Health’ throughout the paper.

7. Regarding the methodology for the fine-mapping analysis, what is the “post-hoc probability for the credible set”? Please clarify in the text. Also, the multi-ancestry fine mapping was performed on SuSiEx, but the single-ancestry fine mapping was conducted with FINEMAP, not SuSiE. Please describe what criteria did the authors consider when selecting the types of fine mapping software.

We have clarified the meaning of this in the methods. FINEMAP was used to complement the findings from SUSIEX as it is a more well-established fine-mapping method, but it cannot perform cross-ancestry fine-mapping.

8. For the PEIZO1-ZFPM1 locus analysis, the posterior inclusion probability of the SAS-specific causal variants should be described in the main text.

Thank you, this has now been included.

9. Regarding the SAS-specific variants, rs563555492, did the author know why the variant is monomorphic in AFR, EUR, and EAS populations? Is there any selection happening at the locus in SAS?

We have explored this, but unfortunately it is challenging to evaluate selection for rare alleles (Fst statistics become unstable at low MAF) and these calculations rely on assumptions on expected heterozygosity rates conforming to Hardy-Weinberg equilibrium. As we have shown, this cohort has high rates of consanguinity and unexpectedly high homozygosity (<https://pubmed.ncbi.nlm.nih.gov/37757828/>), and therefore we feel these analyses require more nuance than is within scope for this paper.

10. The abstract would benefit from improving the clarity. For example, the authors claim their results “illustrates how ancestry-specific genetic variants may be of relevance for the design of clinical reference ranges”. However, there are not substantial results on the design of clinical reference ranges in the main text. The authors also report discovery of genome-wide association signals specific to Gene & Health, but they do not mention that “Gene & Health” is a name of the cohort. The description on the South Asian specific associations on rs563555492 may benefit from including effect size and posterior inclusion probability.

This has been amended.

11. Figures and Table would benefit from further clarification. For example, the error bars in Figure 1B are not defined. The numbers in parentheses for “age at recruitment” in Table 1 is not defined. The abbreviations, for example, EAF (effect allele frequency, in Fig 1B) and PIP (posterior inclusion probability in Fig 2A) are not explicitly defined. In the “variance explained by covariates” plot, the columns are shown in alphabetical order, but the “combined” should be shown at the left-most column or the right-most column.

This has been amended.

12. The sample size of the association study presented in the manuscript is not clearly presented in the main text. In the abstract, the authors claim the “largest” genome-wide association study, but sample size is not provided in the abstract itself. In the introduction, the authors say “Here we report ... a longitudinal genotype-phenotype study comprising ~50,000

individuals ...,” but Table 1 legend says the individuals with both genetic and phenotypic data are up to n=37,352 individuals. The authors should clarify the sample size analyzed in the study across the multiple relevant sections of the manuscript, given that the GWAS discovery from one of the largest South Asian cohorts is the central takeaway message of the manuscript.

This has been amended.

13. Minor points. The phenotype curation process has duplicated descriptions. For example, the selection of 39 traits based on having at least 50% of high-quality measurements are described twice.

This has been amended.

14. Under the “genotyping” subsection in the methods section, biological “sex” should be used instead of “gender”.

This has been amended.

15. Under the “GWAS testing” subsection in the methods section, “IQR 15%” does not represent a “range.”

This has been amended.

16. The sample size of the LD reference panel used in LD-clumping is not described.

This has been added to the manuscript.

17. Under the “genetic correlation” subsection, the authors say “inter-trait heritability” but it should have been written as “genetic correlation.”

This has been amended.

18. Also, the reference for Popcorn used for trans-ancestry genetic correlation is missing.

This has been amended.

Reviewer #2 (Remarks to the Author):

1. The paper description is somewhat unusual as it includes results (Table 1) in the methods section, and discussion in results.

We have amended this.

2. It looks like the study included blood tests that cannot be distinguished as inpatient or outpatient. This distinction is important as inpatient lab tests relate to transitory shifts due to acute disease and are very different from chronic laboratory values obtained in outpatient settings. This is important because they used mean values across multiple dates.

We are grateful to the reviewer for pointing out this important source of heterogeneity. After careful phenotype data quality control (explored in the new supplementary note) we have decided to only use primary care readings for the GWAS, obviating the concern regarding inpatient shifts in blood test values.

3. *The description of the procedures for quality control of the data needs to be clearly described for each laboratory data given their EHR source. It will be important to know fasting status (lipids, glucose), and if participants were taking medications (diabetics taking insulin, and so on). How this compares to procedures used for lab data in published studies (transformation, covariates). For example, HbA1c and glucose should be restricted to individuals without diabetes.*

We have included this information in a new detailed supplementary note which explains our procedures for phenotype data quality control.

4. *A better context on why we should study the genetics of these traits in South Asians needs to be included related to burden of disease or known population genetic findings. Provide some background on the admixture of South Asians in Britain.*

We have amended the introduction to reflect this and included a reference to support this assertion.

5. *It is not clear which of these loci are novel as the comparisons seem to be made with the UK Biobank and not large published GWAS. Replication in South Asians in the UKB would be the preferred step.*

We have used a more conservative definition of ‘novel’ and so have removed these findings from the manuscript as we believe they may have been type 1 error. The signal has fluctuated with additional samples.

6. *Explain why you used 20 PCs given little observed inflation.*

We acknowledge the choice of the number of PCs is a somewhat arbitrary decision, we have followed standard approaches of 20PCs as requested for the covid-19 host genetics initiative consortium (<https://www.ncbi.nlm.nih.gov/pmc/articles/PMC10482689/>), and other international genetics consortia. Given the size of the cohort we have erred on a large number of PCs to model population stratification, and also the much more heterogenous Pakistani population (<https://pubmed.ncbi.nlm.nih.gov/37757828/>). We have included scree plots in the supplement to show that most of the variation in captured by the first few PCs (which are also sufficient to distinguish Bangladeshi from Pakistani ancestry). We would also point out that by using REGENIE, which models and adjusts for the impact of genome-wide variation on each trait in testing individual variants, we gain additional control over population stratification.

7. *A sensitivity analysis using just primary care outpatient labs is needed.*

Thanks to the reviewer’s comments (and similar concerns regarding heterogeneity from reviewer 1) we have changed the analysis plan - now, the primary results presented are from GWAS of primary care results only. Intriguingly, this boosts power despite a slight reduction in sample size, likely due to the reduction in statistical variation due to differences in inpatient vs outpatient tests.

8. Heritability seems too high for some traits, so include the range for heritability in the literature in supplemental table for comparison.

We agree some of the heritability estimates are high and noisy. We expect the noise is due to complex-LD regions such as APOE (for lipid traits). Unfortunately it is not straightforward to compare across heritability estimates across studies and ancestries. As this is not a primary concern of our analysis we have not taken this further.

9. Not sure why the investigators do not use their own data LD for clumping. The overlap with UK Biobank findings should be defined using EUR LD instead of bp range.

We have compared the outcome of using in-sample LD vs using the 1,000 genomes reference LD - as the major purpose of this clumping was for presentation purposes (i.e. to define the top SNP around which a very broad window was drawn) this did not alter results. We have therefore persisted with 1,000 Genomes clumping in the paper.

Regarding the method for testing overlap, we have altered our definition to use a very broad physical window (2MB) either side of the lead variant and a more lenient P value cutoff ($P < 1e-5$) in UKB - this definition is now very conservative to avoid falsely identifying novel loci.

10. For MR-MEGA meta regression, it is important to know if the UK Biobank and your study used the same phenotype transformations and statistical methods because you are combining the summary data.

This has been clarified in the methods.

11. What is the overlap between the UK Biobank SAS participants and those from the Genes & Health study?

We have assessed this question when the Genes & Health cohort was marginally smaller - of the 42,401 Genes & Health volunteers enrolled at that point, based on NHS number, 25 were also in UKB ~ 0.06%. This number is unlikely to have changed significantly with the new Genes & Health volunteers. We have included mention of this in the manuscript.

12. Clarify if the new findings in the Genes & Health study were replicated in the UK Biobank South Asians. It seems that the comparisons were only made for EUR and AFR UKB participants.

We have clarified this in the methods.

13. For the Vit D CUX2 locus, conditional analysis is needed for the published variant in the region.

See above - this finding did not replicate in the extended sample set and so has been dropped from the manuscript.

14. For the PIEZO1 locus, the heterogeneity could be due to trait selection (lack of exclusion of diabetics).

This is an interesting possibility and we thank the reviewer for their thoughtful consideration of why this result has come about. We think that trait selection is unlikely to be a major determinant of the signal we see as a) UK Biobank also did not exclude diabetics either and b) the causal signal we describe is not associated with diabetes or glycaemic control *per se*, but influences HbA1c via its impact on red cell traits.

15. Supplemental figures do not have legends, neither the supplemental tables.

We apologise that these were not clearly visible in the submitted manuscript (they were present but may have been deleted in rendering the pdf for review). We have ensured these are present.

REVIEWER COMMENTS

Reviewer #1 (Remarks to the Author):

In the revised manuscript, Jacobs*, Stow*, Hodgson*, Zöllner*, Samuel* et al. reports genome-wide association mapping of clinically relevant blood biochemistry traits in South Asian ancestry groups by analyzing up to 37,352 individuals in the Genes & Health cohort (G&H) in the UK. The authors addressed many of the concerns raised by both reviewers, and the reviewer appreciated their hard work. The reviewer has a few questions remaining on the methodologies used in the analysis. Overall, The reviewer believes a minor revision should be sufficient for the publication of their study.

1. In the abstract, the authors claim the genetic associations they described "have an impact on the clinical accuracy of commonly used blood tests." However, the main text does not discuss the relevance of the GWAS association and the "clinical accuracy" of blood test panels.

2. In PRIZO1 locus analysis, the authors claim "the only previously reported associations of this SNP [rs563555492] are with red cell traits [...]," but this is not the case. Please see Sun, Q et al. (PubMed ID: 34376796) for reported associations between rs563555492 and HbA1c. They comment on potential erythrocytic mechanisms of genetic control of HbA1c at the PIEZO1 locus. Also, the same variant (rs563555492) is highlighted in variant imputation for non-European populations in the WGS analysis of UKB in Halldorsson et al. (PubMed ID: 35859178). They also provide association results for 459 phenotypes. Please review the literature carefully and contextualize the findings reported in the manuscript in light of works by others.

3. The reviewer appreciates the author's hard work in improving the description of the quality control and revised analysis focusing only on the primary care data. In the reviewed GWAS analysis, the LDSC intercept greater than 1.05 is often considered evidence of systematic bias in the test statistics. Some of the references on this topic include Wendt et al. (PubMed ID: 37268996) and References [27,28,29] therein. Supplementary Table 5 in the revised manuscript indicates that the mean LDSC intercept is 1.08, with a maximum value of 1.15 for MCV. Please confirm if inflation of the association statistics is not an issue in the results provided in the manuscript and/or include a description of the LDSC intercept results (with the values) in the main text.

4. In Fig 2C, did the authors check if there is a statistically significant difference in HbA1c distributions stratified by hard-called dosage count of rs563555492 (and also by the quantile of random glucose)? The statistical significance of the difference in mean and also the difference in distribution (Kolmogorov–Smirnov test) should be assessed.

5. Regarding the Phenotype quality control overview in the Supplementary Text, the curated bespoke code lists and pre-specified clinically plausible ranges should be included in the Supplementary Tables.

6. One of the reviewers previously requested a clean-up of the figures and tables. The authors claimed they had addressed this. However, this is not the case. In the "variance explained by

covariates" plot, the columns are shown in alphabetical order, but the "combined" should be shown at the left-most column or the right-most column. This point is, of course, a minor comment, but the discrepancy between the statement in the rebuttal document and the manuscript package is not nice.

7. The code repository needs clean-up. It has headings like "PICK UP HERE."

Reviewer #2 (Remarks to the Author):

This paper is a great example of genetic studies that apply statistical methods to EHR data without careful curation of the phenotypes, understanding of the clinical datasets and the biology of tested traits. The main findings are based on glycemic traits and I still have main concerns. Because this will be the larger study of South Asians, when summary statistics of this study is published, it will be used as reference for studies of South Asians. Therefore, improving the quality of the research in relation to curated traits is of great importance. In addition, the authors should make an effort to summarize the main new findings and their relevance to the field instead of focus on statistical methods.

The major weakness is the lack of exclusion of diabetics in assessments of glycemic traits. For example, HbA1c values among diabetics in clinical care are driven by patients' glycemic control and treatment, and not necessarily the biology (and genetics) of these traits. The author's insistence in keeping these "outliers" is not appropriate and the assumption that they are confounders is wrong.

The authors missed the reviewer's point for restricting the sample to outpatient lab values when they state that the exclusion is based on trait variance and heterogeneity.

Other issues:

Several questions of Reviewer #1 and #2 are not directly answered in the answer to reviewers, so it is hard to assess if changes were made. I could not find text on the genetic architecture of South Asians or information on fasting status for glucose and lipids, or attempts to retrieve drug treatment that are known to alter the blood levels of these clinical markers. There are standard protocols that use EHR data to account for these issues, and following similar protocols for analyses will be important for comparisons of this study results with published studies.

Limitations section in the discussion are brief and do not address fasting status for some of the biomarkers and the fact that HbA1c analyses included diabetic patients.

REVIEWER COMMENTS

Reviewer #1 (Remarks to the Author):

1. In the abstract, the authors claim the genetic associations they described "have an impact on the clinical accuracy of commonly used blood tests." However, the main text does not discuss the relevance of the GWAS association and the "clinical accuracy" of blood test panels.

We acknowledge this and have removed this from the abstract.

2. In PRIZO1 locus analysis, the authors claim "the only previously reported associations of this SNP [rs563555492] are with red cell traits [...]," but this is not the case. Please see Sun, Q et al. (PubMed ID: 34376796) for reported associations between rs563555492 and HbA1c. They comment on potential erythrocytic mechanisms of genetic control of HbA1c at the PIEZO1 locus. Also, the same variant (rs563555492) is highlighted in variant imputation for non-European populations in the WGS analysis of UKB in Halldorsson et al. (PubMed ID: 35859178). They also provide association results for 459 phenotypes. Please review the literature carefully and contextualize the findings reported in the manuscript in light of works by others.

We thank the reviewer for this suggestion and have amended this section of the paper to include explicit mention of these papers. When we wrote 'red cell traits' in the initial submission we were including the HbA1c association, which we believe is driven by an effect on red cell turnover, but we appreciate this was not clear. We have amended this paragraph as follows:

"Previously-reported associations of this SNP - derived from the subset of 8,149 UKB participants of SAS ancestry – are with red cell traits (haematocrit, haemoglobin, and RBC) and with HbA1c^{12,30}. The effect directions we report in the Genes & Health cohort are in the same effect direction, i.e. the rs563555492_T allele tending to increase haemoglobin and RBC. Whole-genome sequencing data from the first 150,000 UK Biobank genomes independently demonstrated the association of rs563555492 with haemoglobin in SAS ancestry individuals³¹."

3. The reviewer appreciates the author's hard work in improving the description of the quality control and revised analysis focusing only on the primary care data. In the reviewed GWAS analysis, the LDSC intercept greater than 1.05 is often considered evidence of systematic bias in the test statistics. Some of the references on this topic include Wendt et al. (PubMed ID: 37268996) and References [27,28,29] therein. Supplementary Table 5 in the revised manuscript indicates that the mean LDSC intercept is 1.08, with a maximum value of 1.15 for MCV. Please confirm if inflation of the

association statistics is not an issue in the results provided in the manuscript and/or include a description of the LDSC intercept results (with the values) in the main text.

We thank the reviewer for this astute and helpful observation. We would firstly point out that the MR-MEGA analysis is adjusted for genomic control, which is likely to be over-conservative, and so we do already adjust for this inflation which should bias in the direction of the null. In the methods we write “Genomic control statistics (λ) were calculated separately for each cohort and adjusted for in the meta-analysis.”

We have performed further analyses with exclusion of complex LD regions and expanded the sample used to estimate LD scores to $N=1000$ (from $N=450$). Both of these refinements led to an incremental reduction in the LDSC intercepts. We think the slight residual inflation is likely due to the higher relatedness (and consequently extensive IBD sharing) in this cohort (see PMID: 37757828). The key result survives.

4. In Fig 2C, did the authors check if there is a statistically significant difference in HbA1c distributions stratified by hard-called dosage count of rs563555492 (and also by the quantile of random glucose)? The statistical significance of the difference in mean and also the difference in distribution (Kolmogorov–Smirnov test) should be assessed.

Thank you for this suggestion - we have included a formal test of the difference in distributions in the manuscript, and have performed this test among the non-diabetic cohort in response to reviewer 2's comments regarding confounding by diabetic case/control status. Specifically we have written the following:

“Carriers of the T-allele tended to have lower HbA1c levels (rs563555492_{G/G}: median 39.2 mmol/mol, IQR 35.8 – 46.4, rs563555492_{G/T}: median 36.5 mmol/mol, IQR 33.0 – 43.5, rs563555492_{T/T}: median 33.9 mmol/mol, IQR 30.3 – 37.8). We observed the same relationship when restricting to non-diabetic individuals (rs563555492_{G/G}: median 38.0 mmol/mol, IQR 35.0 – 41.8, rs563555492_{G/T}: median 35.3 mmol/mol, IQR 32.3 – 39.3, rs563555492_{T/T}: median 32.3 mmol/mol, IQR 30.1 – 36.1; supplementary figure xxx). In the non-diabetic cohort we performed a statistical comparison of the distribution of mean HbA1c between genotypes, which confirmed the apparent differences between both rs563555492_{G/G} and rs563555492_{G/T} (Kolmogorov-Smirnov test, $P < 2 \times 10^{-16}$), and between rs563555492_{G/T} and rs563555492_{T/T} individuals ($P = 0.003$).”

5. Regarding the Phenotype quality control overview in the Supplementary Text, the curated bespoke code lists and pre-specified clinically plausible ranges should be included in the Supplementary Tables.

We have included these numbers and the codelists used into supplementary table 2. Please note that the ‘limits’ used are to identify obvious outliers due to mis-typed numbers /

technical errors but are deliberately extremely broad so as to capture biological outliers, e.g. very high LDL-cholesterol values in people with Familial Hypercholesterolaemia.

6. One of the reviewers previously requested a clean-up of the figures and tables. The authors claimed they had addressed this. However, this is not the case. In the "variance explained by covariates" plot, the columns are shown in alphabetical order, but the "combined" should be shown at the left-most column or the right-most column. This point is, of course, a minor comment, but the discrepancy between the statement in the rebuttal document and the manuscript package is not nice.

We sincerely apologise for this honest oversight. During the first round of revisions we performed a complete overhaul of the analysis strategy - we re-ran all GWAS and wrote a detailed supplementary note to describe the QC in more detail, which we agree has improved the paper. We overlooked this specific plot in the revisions and have now rectified this error. The plot is copied here for clarity:

7. The code repository needs clean-up. It has headings like "PICK UP HERE."

Apologies for this - we have amended the repository for clarity with better annotation throughout.

Reviewer #2 (Remarks to the Author):

1. The major weakness is the lack of exclusion of diabetics in assessments of glycemic traits. For example, HbA1c values among diabetics in clinical care are driven by patients' glycemic control and treatment, and not necessarily the biology (and genetics) of these

traits. The author's insistence in keeping these "outliers" is not appropriate and the assumption that they are confounders is wrong.

We are very grateful to the reviewer for this suggestion and as a result we have re-run the GWAS of HbA1c stratified by diabetes case/control status. The key result of the paper - the PIEZO1 association with HbA1c - withstands this sensitivity analysis, and in fact strengthens in association with exclusion of diabetics. As the reviewer argues, it does seem that inclusion of the diabetics in the GWAS, despite the additional sample size, does not add a huge amount to statistical power, which we agree is likely due to the increased influence of factors not strictly related to genetics (i.e. treatment). These results are included in the main text and in the supplement.

The regional association at the PIEZO1 locus with HbA1c strengthens following exclusion of diabetics:

We also show that GWAS of HbA1c in non-diabetics uncovers loci which were not significant in the combined analysis:

2. The authors missed the reviewer’s point for restricting the sample to outpatient lab values when they state that the exclusion is based on trait variance and heterogeneity.

We apologise for this and hope we have more closely reflected the reviewer’s concerns in the manuscript, where we now say:

“As blood test readings from hospitalised inpatients are more likely to reflect transient abnormalities in the context of acute illness and so are less likely to be a reflection of steady-state biology, we restricted the dataset to readings from primary care.”

3. Several questions of Reviewer #1 and #2 are not directly answered in the answer to reviewers, so it is hard to assess if changes were made.

We sincerely apologise if we have overlooked further comments - we acknowledge an oversight in not correcting one of the supplementary figures and have now amended this (see response to reviewer 1 above). Some of the comments related to the CUX2 finding which we have dismissed from the manuscript as it did not survive restricting to the primary care data. We believe we have now addressed all the reviewer comments - do please tell us right away if we have not, and we will do our best.

4. I could not find text on the genetic architecture of South Asians or information on fasting status for glucose and lipids, or attempts to retrieve drug treatment that are known to alter the blood levels of these clinical markers. There are standard protocols that use EHR data to account for these issues, and following similar protocols for analyses will be important for comparisons of this study results with published studies.

We have included the following text in the introduction on the genetic architecture of British South Asians:

“British South Asians constitute around 9% of the UK population (UK census 2021, <https://www.ons.gov.uk/>), but around 20% of the UK population with diabetes¹⁰. Both the prevalence of diabetes and the rates of diabetic complications are higher among British South Asians. In addition British South Asians have higher rates of cardiovascular disease (CVD), tend to develop CVD at a younger age, and are at higher risk of early adverse outcomes such as myocardial infarction and stroke¹⁰. The differences in allele frequency and linkage disequilibrium between this population and populations of European Ancestry can help to power the discovery of novel variants associated with traits and diseases which may be missed in Eurocentric analyses¹¹. The genetic architecture of this cohort – which is enriched for consanguinity compared with other large cohorts such as UK Biobank – provides a unique opportunity for understanding the contribution of rare, deleterious coding variation and homozygosity to complex traits and diseases^{12–14}. Understanding the genetic basis of variation in common blood tests (such as glycaemic markers and lipids) in cohorts of South Asian ancestry may therefore help to explain some of these stark healthcare disparities and may shed light on biology of relevance to people of all ancestries^{3,11,15}.”

Regarding fasting status, we acknowledge that this is a limitation of this, and most other studies using routine EHR data, as fasting status is not routinely recorded in these data sources. We would point out that the major finding of the paper is about HbA1c, which is not influenced by fasting. . We have updated the discussion accordingly:

“The major limitation of our study is the lack of an external validation cohort of similar ancestral origin and potential bias due to nonrandom sampling that is an inherent weakness of using routine healthcare data. We chose to not adjust for medication usage as these data are relatively poorly captured in the electronic healthcare records – these records reflect prescription issues, but do not necessarily imply that the participant is taking the medication routinely as prescribed. While we do not adjust for medication usage - such as statins or diabetic medications – in the primary analysis, and this may miss important confounding for some of the GWAS we present, we show in supplementary data that this impact is likely to be relatively minor. We also use a single measure per individual - the mean - to summarise time-varying biomarkers as the outcome for GWAS. This approach has an attractive simplicity, and has the benefit of ‘smoothing’ outlying observations for the individual over time. Of course the major weakness of this approach is that it discards the granularity of how these blood test parameters change over time. This is a question of major importance, but requires bespoke per-trait quality control and consideration of how best to model longitudinal changes in these parameters using EHR data, which is not a trivial task. Furthermore, we are unable to account for the impact of fasting status on lipid measurements or random glucose readings as this information is not readily available in the EHR.”

5. Limitations section in the discussion are brief and do not address fasting status for some of the biomarkers and the fact that HbA1c analyses included diabetic patients.

We acknowledge this helpful point and have revised the discussion to reflect your concern - please see the response to point 4.

REVIEWER COMMENTS

Reviewer #1 (Remarks to the Author):

In the revised manuscript, Jacobs*, Stow*, Hodgson*, Zöllner*, Samuel* et al. reports genome-wide association mapping of blood biochemistry measures in South Asian ancestry groups in the UK by analyzing up to 37,352 individuals in the Genes & Health cohort (G&H). The authors demonstrated substantial efforts in improving the quality of the analysis and clarity of the manuscript. However, the reviewer found it difficult to validate the claims in the point-to-point response documents and remains concerned about the analytical rigor of the primary analysis presented in the manuscript. Overall, the reviewer cannot endorse the publication of the manuscript.

1. The reviewer remains concerned about the analytical rigor presented in the manuscript. The reviewer would like to reiterate the concern of the other reviewer from the previous round: several questions of Reviewer #1 and #2 are not directly addressed in the point-to-point response document. The authors put their answers to the point-to-point response document, but there are instances where it does not directly address the primary concerns of the reviewers. Moreover, the quoted text does not contain page numbers or line numbers, making it extremely challenging to locate the corresponding text in the revised manuscript. Furthermore, new analyses that are not described in the response-to-reviewer document are presented in the manuscript. Together, the reviewer needs to admit that it is challenging to assess the updates in the manuscript in light of the comments from the reviewers.

2. The genomic inflation of the GWAS summary statistics is not fully addressed. In the previous round, one of the reviewers expressed concerns about the large LDSC intercept values. In the response, the authors focused on the quality control of GWAS-meta analysis, one of the downstream analyses, not the GWAS itself, by claiming that “MR-MEGA analysis is adjusted for genomic control.” The adjustment on genomic control in the GWAS meta-analysis does not solve the quality control issue in the GWAS analysis itself.

The authors further claim that “We have performed further analyses with the exclusion of complex LD regions and expanded the sample used to estimate LD scores to N=1000 (from N=450). Both of these refinements led to an incremental reduction in the LDSC intercepts” in their response. However, the reviewer was unable to find the data/results supporting this claim in the manuscript and supplement.

The authors claim, “The key result survives.” The reviewer agrees that the top GWAS hits will likely remain the same, but the quality of the genome-wide summary statistics, which will be the primary results from the manuscript, is of concern. Indeed, the other reviewer also expressed their concerns on the analytical rigor of the GWAS analysis, stating: “Because this will be the larger study of South Asians, when summary statistics of this study is published, it will be used as reference for studies of South Asians. Therefore, improving the quality of the research in relation to curated traits is of great importance.”

It is unfortunate that the authors did not perform sufficient quality control of the primary analysis presented in their manuscript.

3. “cryptic relatedness or population stratification.” The authors claimed that the genomic inflation in the GWAS summary statistics can be attributed to cryptic relatedness or population stratification by stating, “There was some residual inflation (median LDSC intercept 1.09) which may reflect cryptic relatedness or population stratification” in the supplement. If this explanation were to be accurate, we should observe the inflation across all traits. In the results presented in the manuscript, some traits have lower values of LDSC intercept of 1.02, whereas other traits have high values of 1.15. The authors should demonstrate that the genomic inflation is not due to the quality control of the phenotyping and/or lack of adjustment for medication status.

4. Medication adjustment. The authors did not address the requests from the reviewer(s) to consider medications on the phenotypes, which will have non-random effects on the phenotypic values. The authors, instead, claimed that “we show in supplementary data that this impact [medication usage] is likely to be relatively minor.” In the corresponding section of the supplement, the authors presented a non-standard approach to manually inspect the similarity of GWAS association via a Miami plot. As the other reviewer pointed out, there are methods that exist in the field to account for the effects of medication usage. Please see references in this preprint, for example, for more information. Chong et al. Adjusting for medication status in genome-wide association studies. (<https://www.medrxiv.org/content/10.1101/2024.02.19.24303028v1>). Given the substantial genomic inflation in the GWAS summary statistics, it is of critical importance to demonstrate that the lack of medication adjustment did not negatively contribute to the quality of the GWAS results.

5. “falsely lowered HbA1c readings”. Some paragraphs are silently added to the main text, including an interpretation of the impacts of rs563555492 on HbA1c. The authors make claims like “We hypothesize that this variant may therefore distort the relationship between true in vivo glycaemic control versus the biomarker HbA1c, as those with the T allele may have falsely lowered HbA1c for a given average plasma glucose level” and “we reasoned that carriers of the rs563555492T allele may tend to be diagnosed later due to falsely lowered HbA1c readings.” It is possible that the impacts of the genetic variants are localized on HbA1c, not on T2D, and that may influence the risk assessment of T2D based on HbA1c values. However, the HbA1c-lowering effects of the genetic variant do exist, and claiming “falsely lowered HbA1c readings” is not scientifically accurate.

6. Linear regression for the age-of-onset analysis. Related to the previous point, the authors now present a new analysis applying linear regression for the age-of-onset of T2D. For this kind of analysis, a time-to-event model, such as the Cox proportional hazard ratio model accounting for the record length and censoring, would be appropriate rather than linear regression. It is unclear if their “falsely lowered HbA1c readings” claim is supported by appropriate analysis.

Reviewer #2 (Remarks to the Author):

The authors answered all my questions and have appropriated revised the manuscript.

Reviewer responses

Note to reviewers and editors:

Please find our responses to the reviewers' comments in **blue**, the original comments in black, and text quoted directly from the paper in **red**. We have included our responses to the present round of comments and updated responses to the two previous rounds of comments for clarity.

3rd round revisions

Reviewer #1 (Remarks to the Author):

In the revised manuscript, Jacobs*, Stow*, Hodgson*, Zöllner*, Samuel* et al. reports genome-wide association mapping of blood biochemistry measures in South Asian ancestry groups in the UK by analyzing up to 37,352 individuals in the Genes & Health cohort (G&H). The authors demonstrated substantial efforts in improving the quality of the analysis and clarity of the manuscript. However, the reviewer found it difficult to validate the claims in the point-to-point response documents and remains concerned about the analytical rigor of the primary analysis presented in the manuscript. Overall, the reviewer cannot endorse the publication of the manuscript.

1. The reviewer remains concerned about the analytical rigor presented in the manuscript. The reviewer would like to reiterate the concern of the other reviewer from the previous round: several questions of Reviewer #1 and #2 are not directly addressed in the point-to-point response document. The authors put their answers to the point-to-point response document, but there are instances where it does not directly address the primary concerns of the reviewers. Moreover, the quoted text does not contain page numbers or line numbers, making it extremely challenging to locate the corresponding text in the revised manuscript. Furthermore, new analyses that are not described in the response-to-reviewer document are presented in the manuscript. Together, the reviewer needs to admit that it is challenging to assess the updates in the manuscript in light of the comments from the reviewers.

We apologise for the lack of clarity in previous revisions that has led to an additional time burden for the reviewers and are grateful for their continued patience. We have taken several steps to improve the clarity of this document including a flow chart and detailed description of the major changes made. We mention this in the cover letter and copy it below. We have also included line references in the specific point-by-point response to indicate where we have addressed specific comments. To provide an overview of all changes since the initial submission, we have also included responses to all previous comments from both reviewers at the end of this document.

Note that this log of changes is also present in the editor cover letter but is pasted here for clarity. We acknowledge the concerns raised by one reviewer that changes included in previous revisions were implemented somewhat 'silently'. We apologise for this lack of clarity. The major changes from the previous version are as follows:

- Phenotype quality control

- We have included a supplementary figure 1 which summarises the improved phenotype quality control pipeline, copied below:

- The specific steps we have improved/added are as follows:
 - We have refined the codelists used to define laboratory tests in the electronic health record.
 - We have restricted the analysis to individual data points from individuals >18 years old
 - We have improved the stringency of filtering to exclude implausible outlier values in the raw phenotype data using a two-step process
 - i. We have implemented more stringent min/max filters to exclude implausible values, derived using a combination of clinical expertise and (where available) observed ranges and manufacturer ranges for these tests in UK Biobank.
 - ii. We have automatically excluded outliers $>10 \times \text{SD}$ from the mean of each distribution
 - We have accounted for the impact of medications by removing readings occurring after the earliest recorded prescription for the following traits (defined in supplementary table 2):
 - i. Blood lipid measures: statins
 - ii. Markers of glucose homeostasis: anti-hyperglycaemics
 - iii. Vitamin D: vitamin D supplements
 - iv. Folate: folate supplements

- v. Calcium: calcium supplements
 - vi. Ferritin: iron supplements
 - vii. T4 & TSH: levothyroxine
 - Adjustment for age at test and year of test prior to GWAS
- Changes to genetics QC and GWAS
 - Use of TOPMed-r3, an updated imputation panel with significantly improved quality and depth (up to 30% more variants and individuals) compared with the previous version (TOPMed-r2).
 - More stringent SNP-level quality control and individual-level quality control
 - We have regressed out the impacts of age at test and year of test prior to GWAS
 - We have included a slightly larger number of initial traits (n=42 in total) as following detailed further QC we were satisfied that these data were of sufficient quality. Specifically, we now include GWAS summary statistics for the following traits: Aspartate aminotransferase, fasting glucose, and gamma-glutamyl transferase.
- Additional validation steps to explore the functional impact of the *PIEZO1* missense variant:
 - In the updated analysis, the *PIEZO1* variant has a more suggestive impact on plasma glucose (although neither the associations with fasting glucose nor random glucose reached study-wide significance). Although we cannot exclude an effect of the *PIEZO1* variant on plasma glucose levels, the effect on HbA1c is disproportionate compared to other genetic variants strongly linked to plasma glucose levels. In other words, genetic variants linked to plasma glucose levels show a dose-dependent relationship with HbA1c, and the *PIEZO1* variant is a clear outlier in this relationship in that it is associated with a far greater reduction in HbA1c than expected given the associated reduction in plasma glucose.
 - Using observational data, we explore whether the relationship between random or fasting glucose and HbA1c readings (obtained at a similar timepoint) differs among carriers of the *PIEZO1* missense variant. This analysis also supports the concept that this variant is associated with a lower HbA1c for any given level of *in vivo* glycaemia.
 - Finally, we include a time-to-event analysis to explore whether this variant was associated with delayed diagnosis of diabetes, as predicted by our hypothesis. We obtained evidence for delayed diagnosis T2DM diagnosis in people carrying the missense variant using semi-parametric survival models.

2. The genomic inflation of the GWAS summary statistics is not fully addressed. In the previous round, one of the reviewers expressed concerns about the large LDSC intercept values. In the response, the authors focused on the quality control of GWAS-meta analysis, one of the downstream analyses, not the GWAS itself, by claiming that “MR-MEGA analysis is adjusted for genomic control.” The adjustment on genomic control in the GWAS meta-analysis does not solve the quality control issue in the GWAS analysis itself.

Thank you for raising this important point. We have aimed to address this issue through substantially-improved phenotype quality control as described at the beginning of this response document and in supplementary figure 1. These steps have improved measures of genomic inflation, in particular LDSC intercepts are now indistinguishable from 1, implying that the traits with λ_{GC} larger than 1 indicate true polygenicity rather than test statistic inflation due to population stratification (line 84, figure 1B). Note that we have also expanded the size of the panel used to estimate LD scores for LDSC, which is also likely to have improved the LDSC intercepts.

“There was no substantial inflation of test statistics (median λ_{GC} 1.07, range 1.0 - 1.11). Linkage disequilibrium score regression (LDSC) intercepts ranged from 0.95 (Platelets) to 1.02 (Vitamin B12, **figure 1B, supplementary table 3**), falling within the range reported for these traits in UK Biobank and confirming the absence of significant population stratification or other systematic bias³⁴.”

3. The authors further claim that “We have performed further analyses with the exclusion of complex LD regions and expanded the sample used to estimate LD scores to N=1000 (from N=450). Both of these refinements led to an incremental reduction in the LDSC intercepts” in their response. However, the reviewer was unable to find the data/results supporting this claim in the manuscript and supplement.

We apologise for this missing information. During previous rounds of review, in an attempt to address genomic inflation, we tried to optimize each aspect of the analysis pipeline. In the current version of the manuscript, we have now successfully addressed this challenge through a combination of improved quality control for phenotype generation and better imputed genetic variants. The previous response is therefore obsolete and details of how we quantified inflation / performed LDSC can be found in the updated manuscript (line 507-510).

“Test statistic inflation was estimated initially using λ_{GC} (median χ^2 statistic / median χ^2 of a 1-degree of freedom distribution), and subsequently with linkage disequilibrium score regression (LDSC)³⁶ to disentangle inflation due to polygenicity versus population stratification. In the absence of an established LD score reference panel for South Asian ancestry individuals, LD scores were estimated within the Genes & Health cohort.”

4. The authors claim, “The key result survives.” The reviewer agrees that the top GWAS hits will likely remain the same, but the quality of the genome-wide summary statistics, which will be the primary results from the manuscript, is of concern. Indeed, the other reviewer also expressed their concerns on the analytical rigor of the GWAS analysis, stating: “Because this will be the larger study of South Asians, when summary statistics of this study is published, it will be used as reference for studies of South Asians. Therefore, improving the quality of the research in relation to curated traits is of great importance.” It is unfortunate that the authors did not perform sufficient quality control of the primary analysis presented in their manuscript.

We agree that this study will become a resource as it represents the largest GWAS of a range of traits performed in a cohort of South Asian ancestry. We appreciate the importance of pristine quality control to ensure this resource can help to inform further work in an accurate and impactful way. We have carefully addressed these concerns and taken additional steps to ensure rigorous quality control of the underlying data as outlined in the response to point 1. We believe these changes have improved the quality of the genome-wide summary statistics as demonstrated by a) the low genomic inflation (supplementary table 3) and b) the significant overlap with previous GWAS efforts (figures 1-2, supplementary table 5).

5. “cryptic relatedness or population stratification.” The authors claimed that the genomic inflation in the GWAS summary statistics can be attributed to cryptic relatedness or population stratification by stating, “There was some residual inflation (median LDSC intercept 1.09) which may reflect cryptic relatedness or population stratification” in the supplement. If this explanation were to be accurate, we should observe the inflation across all traits. In the results presented in the manuscript, some traits have lower values of LDSC intercept of 1.02, whereas other traits have high values of 1.15. The authors should demonstrate that the genomic inflation is not due to the quality control of the phenotyping and/or lack of adjustment for medication status.

We agree with the reviewer that this was an important limitation of the previous results. We have significantly improved all aspects of the analytic pipeline (see response to point 1) and subsequently we now observe no significant evidence for genomic inflation. Instead, we see LDSC intercepts which are comparable with analysis of these traits in UK Biobank, and which support the concept that the slightly elevated lambda statistics for some traits reflect polygenicity rather than population stratification (or other biases). See figure 1B.

6. Medication adjustment. The authors did not address the requests from the reviewer(s) to consider medications on the phenotypes, which will have non-random effects on the phenotypic values. The authors, instead, claimed that “we show in supplementary data that this impact [medication usage] is likely to be relatively minor.” In the corresponding section of the supplement, the authors presented a non-standard approach to manually inspect the similarity of GWAS association via a Miami plot. As the other reviewer pointed out, there are methods that exist in the field to account for the effects of medication usage. Please see references in this preprint, for example, for more information. Chong et al. Adjusting for medication status in genome-wide association studies. (<https://www.medrxiv.org/content/10.1101/2024.02.19.24303028v1>). Given the substantial genomic inflation in the GWAS summary statistics, it is of critical importance to demonstrate that the lack of medication adjustment did not negatively contribute to the quality of the GWAS results.

We thank the reviewers for raising this important point and acknowledge that medications often have a substantial impact on blood test values in many circumstances, such as statins and LDL cholesterol. We have therefore excluded readings taken after the initiation of specific medications that were likely to affect the specified trait. The affected traits and the number of readings excluded through this process are shown in supplementary table 2. While we are not entirely sure whether this was the major factor leading to decreased genomic inflation measures, the analysed values

now more closely reflect biological values obtained in steady-state, rather than reflecting the impact of medications.

For details please see lines 367 - 378: “To account for the impact of medications of quantitative traits, we first cleaned and curated prescription data from the primary care EHR. We then manually defined a set of commonly-prescribed medications which can influence the traits studied in this paper, and defined the earliest prescription date per-individual per-drug. Rather than perform adjustment of values obtained while ‘on-drug’, we chose the pragmatic approach of restricting to data points obtained prior to drug initiation. While this led to a slight reduction in sample size for some traits and skewed the GWAS cohort towards a healthier population (by excluding readings in individuals on treatment with established disease), this approach led to stricter control of test statistic inflation in GWAS. The following drugs were accounted for: anti-hyperglycaemic agents (insulins, metformin, gliclazide, GLP-1 agonists, SGLT2 inhibitors) for glycaemic traits (HbA1c, random glucose), B12 (for vitamin B12), vitamin D (for serum vitamin D), folate (for serum folate), iron (for serum ferritin), statins (for cholesterol, LDL-C, HDL-C, and triglycerides), thyroxine (for T4 and TSH). The list of traits adjusted for medications is shown in the supplement (**supplementary table 2**).”

We thank the reviewer for pointing out this elegant recent work that came out while our work was under review. While the described approach is clearly interesting and effective, a major limitation of the medication data we have available is that we have records of prescriptions being issued, but we have no robust way of determining whether participants took this medication as prescribed. Given that a major strength of our data is its richness (both in terms of sheer size but also longitudinal breadth), we have chosen the conservative approach of discarding blood test values obtained after the initiation of specific medications for specific traits. While this approach discards some data, (the exact number of readings lost for each trait is outlined in supplementary table 2), we hope the reviewers will agree that this approach strikes an appropriate balance between sample size and specificity.

7. “falsely lowered HbA1c readings”. Some paragraphs are silently added to the main text, including an interpretation of the impacts of rs563555492 on HbA1c. The authors make claims like “We hypothesize that this variant may therefore distort the relationship between true in vivo glycaemic control versus the biomarker HbA1c, as those with the T allele may have falsely lowered HbA1c for a given average plasma glucose level” and “we reasoned that carriers of the rs563555492T allele may tend to be diagnosed later due to falsely lowered HbA1c readings.” It is possible that the impacts of the genetic variants are localized on HbA1c, not on T2D, and that may influence the risk assessment of T2D based on HbA1c values. However, the HbA1c-lowering effects of the genetic variant do exist, and claiming “falsely lowered HbA1c readings” is not scientifically accurate.

We apologise for the lack of transparency regarding changes during the revision process and now provided a detailed response to previous comments below, as well as a comprehensive log of all changes in response to point 1 above. We agree with the reviewer that specific care needs to be taken when interpreting the role of rs563555492 in HbA1c biology and subsequent impact on the

diagnosis of type 2 diabetes. We have carefully rephrased the section to acknowledge the fact that while the variant may likely act via non-glycaemic pathways on T2DM, and hence is not a causal factor, it may well interfere with current screening procedures to identify people at high risk of T2DM that may eventually delay diagnosis and appropriate treatment (see lines 244-251).

8. Linear regression for the age-of-onset analysis. Related to the previous point, the authors now present a new analysis applying linear regression for the age-of-onset of T2D. For this kind of analysis, a time-to-event model, such as the Cox proportional hazard ratio model accounting for the record length and censoring, would be appropriate rather than linear regression. It is unclear if their “falsely lowered HbA1c readings” claim is supported by appropriate analysis.

We thank the reviewer for this valuable comment and have implemented the suggested analysis. This supports our previous conclusions using a more appropriate statistical test. We have amended this section to include time-to-event analysis (figure 5F, lines 244 onwards, and online methods line 560 onwards):

“Given the HbA1c is widely used to diagnose diabetes, we reasoned that carriers of the rs563555492_T allele may tend to be diagnosed later due to disproportionately lowered HbA1c readings for any given level of average blood glucose. Of the 51,032 individuals included in this analysis (median 24.4 years of follow-up, total follow-up 1.3 million years), the median age at earliest T2DM diagnosis was 46 years old for carriers of the rs563555492_T allele (N = 862) and 45 years old for non-carriers (N = 12,040). We confirmed the impact of rs563555492_T on delaying T2DM diagnosis using a flexible parametric survival model²⁴, which showed a marginal estimate of median survival for rs563555492_T carriers 61.8 years vs 60.2 years for non-carriers, i.e. a delay, on average, of 1.6 years in those with the rs563555492_T allele (figure 5F).”

2nd round revisions

In the revised manuscript, Jacobs*, Stow*, Hodgson*, Zöllner*, Samuel* et al. reports genome-wide association mapping of clinically relevant blood biochemistry traits in South Asian ancestry groups by analyzing up to 37,352 individuals in the Genes & Health cohort (G&H) in the UK. The authors addressed many of the concerns raised by both reviewers, and the reviewer appreciated their hard work. The reviewer has a few questions remaining on the methodologies used in the analysis. Overall, The reviewer believes a minor revision should be sufficient for the publication of their study.

1. In the abstract, the authors claim the genetic associations they described "have an impact on the clinical accuracy of commonly used blood tests." However, the main text does not discuss the relevance of the GWAS association and the "clinical accuracy" of blood test panels.

We have amended this as follows (line 37):

“Our results shed light on the genetic basis of clinically-relevant quantitative traits in an under-represented population, and emphasise the importance of ancestral diversity in genetic studies.”

2. In PRIZO1 locus analysis, the authors claim "the only previously reported associations of this SNP [rs563555492] are with red cell traits [...]," but this is not the case. Please see Sun, Q et al. (PubMed ID: 34376796) for reported associations between rs563555492 and HbA1c. They comment on potential erythrocytic mechanisms of genetic control of HbA1c at the PIEZO1 locus. Also, the same variant (rs563555492) is highlighted in variant imputation for non-European populations in the WGS analysis of UKB in Halldorsson et al. (PubMed ID: 35859178). They also provide association results for 459 phenotypes. Please review the literature carefully and contextualize the findings reported in the manuscript in light of works by others.

This is now included in line 190 - 202:

“Previously-reported associations of this SNP – derived from the subset of 8,149 UKB participants of SAS ancestry – are also with red cell traits (haematocrit, haemoglobin, and RBC) and with HbA1c^{11,39}. The effect directions we report in the Genes & Health cohort are in the same effect direction, i.e. the rs563555492_T allele was associated with increased haemoglobin and RBC, implying the rare (T) allele may protect against anaemia. Whole-genome sequencing data from the first 150,000 UK Biobank genomes independently demonstrated the association of rs563555492 with haemoglobin in SAS ancestry individuals⁴⁰. Although we excluded readings from individuals taking anti-hyperglycaemic medications, and so our results were largely obtained from controls or people who at the time of their blood tests were not diagnosed with diabetes, we sought to ensure that these findings were not confounded by disease status. To do so we conducted stratified GWAS of HbA1c in Genes & Health participants without diabetes (N=24,783), excluding individuals diagnosed at any point during their electronic healthcare record follow-up. The strong association of rs563555492 with HbA1c persisted, and in fact strengthened in statistical significance in the non-diabetic cohort (Non-diabetics: $\beta = -0.64$, $P = 1.2 \times 10^{-181}$; beta orientated to the rs563555392_T allele).”

3. The reviewer appreciates the author's hard work in improving the description of the quality control and revised analysis focusing only on the primary care data. In the reviewed GWAS analysis, the LDSC intercept greater than 1.05 is often considered evidence of systematic bias in the test statistics. Some of the references on this topic include Wendt et al. (PubMed ID: 37268996) and References [27,28,29] therein. Supplementary Table 5 in the revised manuscript indicates that the mean LDSC intercept is 1.08, with a maximum value of 1.15 for MCV. Please confirm if inflation of the association statistics is not an issue in the results provided in the manuscript and/or include a description of the LDSC intercept results (with the values) in the main text.

Please see response to reviewer comment number 2, round 3 reviews.

4. In Fig 2C, did the authors check if there is a statistically significant difference in HbA1c distributions stratified by hard-called dosage count of rs563555492 (and also by the quantile of random glucose)? The statistical significance of the difference in mean and also the difference in distribution (Kolmogorov–Smirnov test) should be assessed.

We had included this in a previous version of the manuscript but have now omitted this in favour of other analyses aimed at validating this finding (see lines 204 onwards). The K-S test was highly significant, but we do not feel this would add anything further to the manuscript as is.

5. Regarding the Phenotype quality control overview in the Supplementary Text, the curated bespoke code lists and pre-specified clinically plausible ranges should be included in the Supplementary Tables.

These are included in supplementary table 2.

6. One of the reviewers previously requested a clean-up of the figures and tables. The authors claimed they had addressed this. However, this is not the case. In the "variance explained by covariates" plot, the columns are shown in alphabetical order, but the "combined" should be shown at the left-most column or the right-most column. This point is, of course, a minor comment, but the discrepancy between the statement in the rebuttal document and the manuscript package is not nice.

We have re-made all figures and tables, and hope that the reviewers will agree they are crisper and easier to understand.

7. The code repository needs clean-up. It has headings like "PICK UP HERE."

This has been amended.

Reviewer #2 (Remarks to the Author):

8. This paper is a great example of genetic studies that apply statistical methods to EHR data without careful curation of the phenotypes, understanding of the clinical datasets and the biology of tested traits. The main findings are based on glycemc traits and I still have main concerns. Because this will be the larger study of South Asians, when summary statistics of this study is published, it will be used as reference for studies of South Asians. Therefore, improving the quality of the research in relation to curated traits is of great importance. In addition, the authors should make an effort to summarize the main new findings and their relevance to the field instead of focus on statistical methods.

We hope we have addressed this comment with some further high-level context of the findings (i.e. line 99 - 105):

"We replicated several well-known associations across all traits tested, such as for LDL-cholesterol (*PCSK9*, *APOB*, *LDLR*, *HMGCR*, and *APOE*²), vitamin D (*GC*)³⁵, creatinine (*GATM*)³⁶, and HbA1c (*HHEX*, *GCK*, and *TCF7L2*; figure 1C)⁶. We [BJ1] detected some associations where the SNP was common (MAF \geq 0.01) in people of South Asian ancestry but ultra-rare (MAF $<$

0.0001) in other ancestral populations, and therefore likely to be missed in genetic analysis restricted to European ancestry. For example, we replicated the previously-reported South Asian ancestry-specific associations between missense variants in *LIPG* with HDL-C², in *IL7*¹¹ with lymphocyte count, and in *GPI1B*³⁷ with platelet count (supplementary table 4).”

9. The major weakness is the lack of exclusion of diabetics in assessments of glycemc traits. For example, HbA1c values among diabetics in clinical care are driven by patients’ glycemc control and treatment, and not necessarily the biology (and genetics) of these traits. The author’s insistence in keeping these “outliers” is not appropriate and the assumption that they are confounders is wrong.

We have included GWAS of HbA1c stratified by T2DM status as a validation for the *PIEZO1* finding (line 198 - 202):

“To do so we conducted stratified GWAS of HbA1c in Genes & Health participants without diabetes (N=24,783), excluding individuals diagnosed at any point during their electronic healthcare record follow-up. The strong association of rs563555492 with HbA1c persisted, and in fact strengthened in statistical significance in the non-diabetic cohort (Non-diabetics: $\beta = -0.64$, $P = 1.2 \times 10^{-181}$; beta orientated to the rs563355392_T allele).”

In addition we have excluded readings taken on or after the initiation of anti-hyperglycaemic medications, so the GWAS of glycaemic traits, although they do include diabetics, should be restricted to the ‘untreated’ phase and therefore should not be confounded by medication.

10. The authors missed the reviewer’s point for restricting the sample to outpatient lab values when they state that the exclusion is based on trait variance and heterogeneity.

The main analyses are all performed exclusively with primary care data now. We think we broadly agree with the reviewer on why it is challenging to integrated inpatient data, and have written the following (lines 380 -384):

“Blood test results for some assays in hospitalised inpatients are more likely to reflect transient abnormalities related to acute illness and so are less likely to be a reflection of steady-state biology. While the overall population mean values for primary and secondary care traits were highly correlated (supplementary figure 2), there was substantially greater heterogeneity in the secondary care data (supplementary figure 3). We therefore restricted the dataset to readings from primary care.”

11. Several questions of Reviewer #1 and #2 are not directly answered in the answer to reviewers, so it is hard to assess if changes were made. I could not find text on the genetic architecture of South Asians or information on fasting status for glucose and lipids, or attempts to retrieve drug treatment that are known to alter the blood levels of these clinical markers. There are standard

protocols that use EHR data to account for these issues, and following similar protocols for analyses will be important for comparisons of this study results with published studies.

We have included mention of these points as follows:

- Genetic architecture of South Asians: we have included the following context in the introduction (line 54 onwards): “British South Asians constitute around 9% of the UK population (UK census 2021, <https://www.ons.gov.uk/>), but around 20% of the UK population with diabetes¹⁰. Both the prevalence of diabetes and the rates of diabetic complications are higher among British South Asians. In addition British South Asians have higher rates of cardiovascular disease (CVD), tend to develop CVD at a younger age, and are at higher risk of early adverse outcomes such as myocardial infarction and stroke¹⁰. The differences in allele frequency and linkage disequilibrium between this population and populations of European ancestry can help to power the discovery of novel variants associated with traits and diseases which may be missed in Eurocentric analyses¹¹. The genetic architecture of this cohort – which is enriched for consanguinity compared with other large cohorts such as UK Biobank – provides a unique opportunity for understanding the contribution of rare, deleterious coding variation and homozygosity to complex traits and diseases^{12–14}. Understanding the genetic basis of variation in common blood tests (such as glycaemic markers and lipids) in cohorts of South Asian ancestry may therefore help to explain some of these stark healthcare disparities and may shed light on biology of relevance to people of all ancestries^{3,11,15}.”
- Fasting status: we have made explicit in the discussion that the lack of fasting status for lipids is a weakness, but include a GWAS of fasting glucose (line 320).
- Re drug treatment: see point 6 response, round 3 reviews

12. Limitations section in the discussion are brief and do not address fasting status for some of the biomarkers and the fact that HbA1c analyses included diabetic patients.

Please see line 320 re fasting status and round 2 review response 9.

“Other limitations of our study regarding the phenotypes examined include the relatively small number of traits tested and the lack of clinical context for some tests (e.g. we cannot determine whether the observed lipid biomarkers were measured in a fasted state).”

1st round revisions

Reviewer #1 (Remarks to the Author):

1. The authors have already shown great efforts to demonstrate the quality of their phenotypic measurements. Given the non-random sampling of the phenotypes, I believe there is still some additional analysis and quality control needs to be done. What is the distribution of the number of blood biochemistry measurements across individuals? If some individuals have an extremely large number of measurements, it may perhaps reflect the underlying disease status that necessitated the repeated measurements of the trait and perhaps such extreme outlier individuals should be excluded from the association analysis. Similarly, did the author assess/adjust for the length of

the record length (number of the event and years of the record) when conducting association analysis? The authors showed that the measurements from each individual across different time points are highly correlated ($p < 10^{-10}$, indicating the non-zero correlation of the measurements), but do not show whether the intra-individual correlation is higher than inter-individual correlations (perhaps permutation test p-value would be appropriate).

We have previously shown the reviewer these plots and demonstrated the variability in number of readings. However, we think that there is a danger of introducing collider bias by excluding diseased individuals or adjust for the number of readings (which may be correlated with disease status). We have omitted this from the supplement. For the *PIEZO1* finding however we do prove replication in the non-diabetic cohort to provide further reassurance that this finding was not driven solely by people with diagnosed T2DM (line 200): “The strong association of rs563555492 with HbA1c persisted, and in fact strengthened in statistical significance in the non-diabetic cohort (Non-diabetics: $\beta = -0.64$, $P = 1.2 \times 10^{-181}$; beta orientated to the rs563355392_T allele).”

2. The discovery of the genome-wide association signals that was not previously reported in the literature is one of the strengths of the study. The authors did a great job in evaluating whether the association signals were replicated in other cohorts, such as UK Biobank. However, I am puzzled to see the rs7314285(CUX2)-VitD association was in the African and European ancestry groups in UK Biobank (UKB), but not in South Asian and East Asian ancestry groups in UK Biobank. Is it simply because the simplistic description of “South Asian” in UKB and G&H does not reflect the heterogeneity in the ancestry groups? Or, is it stemming from the fact that G&H collects blood biochemistry from clinical settings whereas UKB measured biochemistry from blood samples collected at the assessment center? Please provide more description on why or why there is no replication of the findings on the rs7314285(CUX2)-VitD association in UKB’s “South Asian” cohort.

Please note that the CUX2 finding has been removed from the paper as we instituted a more stringent study-wide statistical significance threshold. We wonder if this result could represent a false positive from having previously used a more lenient threshold ($P < 5 \times 10^{-8}$).

3. Related to the point above, are the two sets of summary statistics from UKB and G&H directly comparable in terms of the phenotypic transformation? Given the skew of the distributions, it is common to apply log-transformation of the phenotype before conducting the GWAS analysis. I wonder if the lack of the replication of some of their findings can also be explained by the difference in phenotypic transformation.

Rank transformation destroys the original scale of the trait, and so analyses which incorporate effect sizes across such studies should be interpreted with a degree of caution. However as we show in figure 2 there is a positive genetic correlation across all traits between UKB EUR and Genes & Health, showing that the genetic bases of these traits are highly comparable even though the phenotype transformations are not identical.

4. At the CUX2 locus, authors report the moderate linkage of two SNPs (rs7314285 and rs73413596) in South Asian. What are the LD statistics in European individuals? Also, did the authors confirm if the “novel” rs7314285(CUX2)-VitD association is significant after applying conditional analysis on rs73413596?

Please see response to comment 2. This finding has been omitted from the revised manuscript as it no longer achieved study-wide significance.

5. In the description of the rs7314285(CUX2)-VitD association, the authors mention that the variant is also associated with sex hormone binding globulin and testosterone levels in UK Biobank. Are those fine-mapped causal variants? What is the relevance of the pleiotropic associations in interpreting the reported “novel” associations? Please clarify in the main text.

Please see response to comment 2. This finding has been omitted from the revised manuscript as it no longer achieved study-wide significance.

6. The use of the term “novel” may not be appropriate. Associations previously not reported in the literature (or in UK Biobank) would be more appropriate.

We have omitted the use of the word novel.

7. Regarding the methodology for the fine-mapping analysis, what is the “post-hoc probability for the credible set”? Please clarify in the text. Also, the multi-ancestry fine mapping was performed on SuSiEx, but the single-ancestry fine mapping was conducted with FINEMAP, not SuSiE. Please describe what criteria did the authors consider when selecting the types of fine mapping software.

We have clarified this in line 161: “One of these four causal signals was specific to persons of SAS ancestry, i.e. the post-hoc probability of being causal was >0.8 in both Genes & Health and SAS-ancestry UK Biobank, but not in other ancestral groups. This variant - rs72807421 (chr16:88524059:C>T) – was also a SAS-specific causal variant for total bilirubin and for haemoglobin.”

Regarding the choice of fine-mapping software, SuSiEx has the main advantage of being able to perform multi-ancestry fine-mapping, whereas FINEMAP is a more well-established programme that works with a single ancestry only. We could have run SusiEX for the single-ancestry fine mapping but chose to use FINEMAP as it is better known to the community. Please note we also use COJO for conditional analysis and perform conditional analysis in REGENIE, so prove using 3 orthogonal approaches that this variant is driving the association.

8. For the PEIZO1-ZFPM1 locus analysis, the posterior inclusion probability of the SAS-specific causal variants should be described in the main text.

This is included now (line 177).

9. Regarding the SAS-specific variants, rs563555492, did the author know why the variant is monomorphic in AFR, EUR, and EAS populations? Is there any selection happening at the locus in SAS?

We are unsure of this, and we think further work to quantify the selection at this locus would be of interest but is beyond this paper, as methods for estimating F_{st} in populations such as this with high homozygosity rates are not well-developed.

10. The abstract would benefit from improving the clarity. For example, the authors claim their results “illustrates how ancestry-specific genetic variants may be of relevance for the design of clinical reference ranges”. However, there are not substantial results on the design of clinical reference ranges in the main text. The authors also report discovery of genome-wide association signals specific to Gene & Health, but they do not mention that “Gene & Health” is a name of the cohort. The description on the South Asian specific associations on rs563555492 may benefit from including effect size and posterior inclusion probability.

We have amended the abstract as follows, but have omitted the effect size itself in favour of the crude HbA1c values per genotype, as the effect size is not directly interpretable due to the RINT transformation:

“Understanding the genetic basis of blood tests routinely acquired in clinical care can provide insights into several aspects of human physiology and disease. Previous efforts have almost exclusively focussed on individuals of European descent, severely limiting generalizability to ancestrally-diverse populations. Here we describe the largest genome-wide association study of quantitative traits in a cohort of South Asian ancestry (up to $N = 38,224$). We tested the association of ~4.8 million autosomal, common (minor allele frequency [MAF] > 1%) Single Nucleotide Polymorphisms (SNPs). We defined 42 quantitative blood test traits from Electronic Healthcare Records (EHRs) of ~50,000 British Bangladeshi and British Pakistani adults. By performing multi-ancestry meta-analysis and fine mapping using data from UK Biobank, we demonstrate an ancestry-specific causal variant within the *PIEZO1* locus which was associated with alterations in red cell traits and glycated haemoglobin. Conditional analysis and within-ancestry fine mapping confirmed that this signal is driven by a missense variant - rs563555492 – which is common in South Asian ancestries (MAF 3.9%) but ultra-rare in other ancestries. Carriers of the T allele tended to have lower mean HbA1c values (Mean HbA1c: rs563555492_{G/G} 40.8 mmol/mol [SD = 10.7]; rs563555492_{G/T} 38.2 mmol/mol [SD=11.1]; rs563555492_{T/T} 34.0 mmol/mol [SD=6.8]), lower HbA1c values for a given level of random or fasting glucose, and later Type 2 Diabetes Mellitus diagnosis. Our results shed light on the genetic basis of clinically-relevant quantitative traits in an under-represented population, and emphasise the importance of ancestral diversity in genetic studies.”

11. Figures and Table would benefit from further clarification. For example, the error bars in Figure 1B are not defined. The numbers in parentheses for “age at recruitment” in Table 1 is not defined.

The abbreviations, for example, EAF (effect allele frequency, in Fig 1B) and PIP (posterior inclusion probability in Fig 2A) are not explicitly defined. In the “variance explained by covariates” plot, the columns are shown in alphabetical order, but the “combined” should be shown at the left-most column or the right-most column.

We have substantially updated all figures and tables.

12. The sample size of the association study presented in the manuscript is not clearly presented in the main text. In the abstract, the authors claim the “largest” genome-wide association study, but sample size is not provided in the abstract itself. In the introduction, the authors say “Here we report ... a longitudinal genotype-phenotype study comprising ~50,000 individuals ...,” but Table 1 legend says the individuals with both genetic and phenotypic data are up to $n=37,352$ individuals. The authors should clarify the sample size analyzed in the study across the multiple relevant sections of the manuscript, given that the GWAS discovery from one of the largest South Asian cohorts is the central takeaway message of the manuscript.

We have made this explicit in supplementary table 2.

13. Minor points. The phenotype curation process has duplicated descriptions. For example, the selection of 39 traits based on having at least 50% of high-quality measurements are described twice. Under the “genotyping” subsection in the methods section, biological “sex” should be used instead of “gender”. Under the “GWAS testing” subsection in the methods section, “IQR 15%” does not represent a “range.” The sample size of the LD reference panel used in LD-clumping is not described. Under the “genetic correlation” subsection, the authors say “inter-trait heritability” but it should have been written as “genetic correlation.” Also, the reference for Popcorn used for trans-ancestry genetic correlation is missing.

This is amended.

Reviewer #2 (Remarks to the Author):

14. The paper description is somewhat unusual as it includes results (Table 1) in the methods section, and discussion in results.

We have amended the structure to more closely reflect what we hope will be the published version (introduction, results, discussion, online methods).

15. It looks like the study included blood tests that cannot be distinguished as inpatient or outpatient. This distinction is important as inpatient lab tests relate to transitory shifts due to acute disease and are very different from chronic laboratory values obtained in outpatient settings. This is important because they used mean values across multiple dates.

We have now restricted the analysis to primary care only.

16. The description of the procedures for quality control of the data needs to be clearly described for each laboratory data given their EHR source. It will be important to know fasting status (lipids, glucose), and if participants were taking medications (diabetics taking insulin, and so on). How this compares to procedures used for lab data in published studies (transformation, covariates). For example, HbA1c and glucose should be restricted to individuals without diabetes.

We have detailed this more explicitly in the methods and in supplementary figure 1.

17. A better context on why we should study the genetics of these traits in South Asians needs to be included related to burden of disease or known population genetic findings. Provide some background on the admixture of South Asians in Britain.

Please see response to point 11, round 2 reviews.

5. It is not clear which of these loci are novel as the comparisons seem to be made with the UK Biobank and not large published GWAS. Replication in South Asians in the UKB would be the preferred step.

6. We now make explicit in line 105 that none of these loci are novel:

“For the 29 traits which overlapped with UK Biobank (UKB), all of the detected signals in the Genes & Health cohort were within regions harbouring suggestive associations in UKB (i.e. within +/- 1MB of a variant with $P < 1 \times 10^{-5}$ in at least one ancestral group in UKB).”

7. Explain why you used 20 PCs given little observed inflation.

This is an arbitrary choice and in our current GWAS has clearly provided adequate control of inflation. We had previously included scree plots in the supplement but have now removed these as they did not provide much additional information in our view.

8. A sensitivity analysis using just primary care outpatient labs is needed.

The paper now exclusively uses primary care data.

9. Heritability seems too high for some traits, so include the range for heritability in the literature in supplemental table for comparison.

We have removed the heritability estimation as we did not feel this was adding much insight to the paper.

10. Not sure why the investigators do not use their own data LD for clumping. The overlap with UK Biobank findings should be defined using EUR LD instead of bp range.

We use Genes & Health LD to identify independent lead SNPs (supplementary table 4). We believe the 2-MB positional range is sufficient rather than an LD-based window for defining novelty. It is likely a little conservative, but we are erring on the side of conservativeness as we do not want to report false positive 'novel' associations.

11. For MR-MEGA meta regression, it is important to know if the UK Biobank and your study used the same phenotype transformations and statistical methods because you are combining the summary data.

Please see response to point 3 above.

12. What is the overlap between the UK Biobank SAS participants and those from the Genes & Health study?

This is included in line 478: "Based on NHS numbers, approximately 0.05% of Genes & Health participants have also taken part in UK Biobank, and so participant overlap is unlikely to introduce significant bias."

13. Clarify if the new findings in the Genes & Health study were replicated in the UK Biobank South Asians. It seems that the comparisons were only made for EUR and AFR UKB participants.

All GWAS were systematically compared with UKB for the 29 overlapping traits.

14. For the Vit D CUX2 locus, conditional analysis is needed for the published variant in the region.

This finding has been removed due to the increased stringency of type 1 error control.

15. For the PIEZO1 locus, the heterogeneity could be due to trait selection (lack of exclusion of diabetics).

This finding increased in strength in the analysis of non-diabetics.

16. Supplemental figures do not have legends, neither the supplemental tables.

These have been amended.

REVIEWER COMMENTS

Reviewer #1 (Remarks to the Author):

Jacobs*, Stow*, Hodgson*, Zöllner*, Samuel* et al. substantially improved the quality of genome-wide associations for 42 traits in South Asian ancestry groups in the Genes & Health cohort (G&H) cohort in the UK. The quality of the association results has improved. The reviewer's remaining concern is the multi-ancestry fine-mapping results presented in Figure 4 and the interpretation of the results. Overall, the reviewer cannot recommend the publication of the current version of the manuscript.

1. The multi-ancestry fine mapping. The reviewer is confused on how to interpret the multi-ancestry fine-mapping results. The results in Figure 4A do not match the rest of the manuscript. Specifically, the PIEZO1 missense variant (rs563555492, chr16:88716656:G>T), discussed extensively in Figure 5 as the putative causal variant, is not prioritized in multi-ancestry fine mapping. The variant is not included in the four distinct causal signals in Supplementary Table 8. The authors themselves cast doubt on the results by saying, "We considered whether it may not be the true causal variant but may in fact be tagging a coding variant" in lines 173-174. However, it is unlikely that "tagging" is the case. The four fine-mapped variants and LD with the missense variant (chr16:88716656:G>T) are chr16:88789676:C>G (R2=0.02), chr16:88524059:C>T (R2=0.27), chr16:88717706:T>C (R2=0.07), and chr16:88991721:C>T (R2=0.00), according to the data presented in Supplementary Table 9. The authors should drop the multi-ancestry fine-mapping results or guide the readers on how to interpret the discrepancy between the two sets of results from fine-mapping analysis.

2. Figure 4B. Figure 4B does not match with the main text. There is a description on pleiotropic association of a variant to bilirubin and Hb in line 163. However, the main text reports the results for rs72807421 (chr16:88524059:C>T), but the figure reports the results for the PIEZO1 missense variant (rs563555492, chr16:88716656:G>T).

3. The same variant is referred to by multiple nomenclatures, including "chr:pos," "chr:pos-A1", "chr:pos:A2>A1", or rsID. The same applies to locus names (PIEZO1 locus, CYBA-PIEZO1 locus). It is confusing. Please be consistent across main text (for example, rs563555492_T and PIEZO1 locus) and main figure panels. Related to this, please provide a genome browser view for PIEZO1 locus. There are references to gene symbols in the regions, such as "an intronic SNP lying between exons 4 and 5 of ZFPM1" (line 164-165), and a browser view would help contextualize the claims.

4. Ancestry-specific variants vs. ancestry-specific effects. The authors should not claim ancestry-specific genetic variant effects when the lack of observed associations in non-SAS populations is likely to be explained by the lack of power due to the low allele frequency of the variant in non-SAS populations. For example, the "selective effect" in line 169 is problematic. Also, the "Heterogeneity of effect size" in line 295 should be revised.

5. Figure 5E. The reviewer has a hard time understanding the point of this analysis. The authors stratified individuals based on the genotype at rs563555492, constructed a predictive model of

HbA1c using G/G carriers, and predicted HbA1c values for T/T or G/T carriers. The prediction is off, as expected because the variant rs56355492 has associations with HbA1c. What do we learn from this exercise beyond the fact that the variant is associated with HbA1c (which has already been reported elsewhere)?

6. The corresponding hypothesis in lines 218-220 better fits in discussion. The "true" in vivo glycemic control is not observable and cannot be validated based on the results presented in the manuscript.

7. The "Design of reference ranges" discussion in line 304 is an overstatement. There are no results on "reference ranges" in the results.

Review response

We thank the reviewer for their constructive comments on the manuscript and have addressed their outstanding concerns as follows.

1. The multi-ancestry fine mapping. The reviewer is confused on how to interpret the multi-ancestry fine-mapping results. The results in Figure 4A do not match the rest of the manuscript. Specifically, the PIEZO1 missense variant (rs563555492, chr16:88716656:G>T), discussed extensively in Figure 5 as the putative causal variant, is not prioritized in multi-ancestry fine mapping. The variant is not included in the four distinct causal signals in Supplementary Table 8. The authors themselves cast doubt on the results by saying, "We considered whether it may not be the true causal variant but may in fact be tagging a coding variant" in lines 173-174. However, it is unlikely that "tagging" is the case. The four fine-mapped variants and LD with the missense variant (chr16:88716656:G:T) are chr16:88789676:C:G ($R^2=0.02$), chr16:88524059:C:T ($R^2=0.27$), chr16:88717706:T:C ($R^2=0.07$), and chr16:88991721:C:T ($R^2=0.00$), according to the data presented in Supplementary Table 9. The authors should drop the multi-ancestry fine-mapping results or guide the readers on how to interpret the discrepancy between the two sets of results from fine-mapping analysis.

We thank the reviewer for this comment and apologise for the lack of clarity. In the previous iteration of the manuscript, we only performed fine-mapping with variants which were common ($MAF > 0.01$) in all ancestries tested, and so the causal signals (and credible sets) refer to the set of variants which was common ($MAF > 0.01$) in all cohorts. It follows that the causal signals identified could be better explained by a variant which was not included in the fine mapping due to low MAF in a cohort, as was the case for rs563555492, which was not included in the cross-ancestry fine mapping due to its low allele frequency in UKB Europeans. Specifically, previously we had excluded variants with an overall allele frequency of < 0.01 in UKB, which meant that rs563555492 was not included and hence could not be prioritised.

We acknowledge this was not clear. To rectify this, we have repeated the fine-mapping using variants from UKB which were common ($MAF > 0.01$) in at least two of the cohorts tested (i.e. rs563555492 was retained as the MAF is 0.035 in UKB SAS individuals). Using this approach, fine mapping of this locus revealed six distinct signals, of which rs563555492 was the likely causal variant for the causal signal with evidence for SAS-specificity (i.e. a post-hoc probability of being causal of >0.8 in both Genes & Health and UKB South Asian ancestry persons, but not in other UKB ancestries).

To further address the reviewer's concern and ensure consistency in the manuscript, we have re-run all of the fine-mapping analyses using this approach (which is likely more sensitive to 'ancestry-specific effects' as it allowed us to retain variants which were common in both SAS cohorts but not others). The updated results are in supplementary table 8.

2. Figure 4B. Figure 4B does not match with the main text. There is a description on pleiotropic association of a variant to bilirubin and Hb in line 163. However, the main text reports the results for rs72807421 (chr16:88524059:C>T), but the figure reports the results for the PIEZO1 missense variant (rs563555492, chr16:88716656:G>T).

We have amended the text such that both the figure and text now focus on chr16-88716656-G-T. Both of these variants show pleiotropic associations, but we appreciate that it was unclear for the reader to discuss one variant in the text and show another in the figure.

3. The same variant is referred to by multiple nomenclatures, including "chr:pos," "chr:pos-A1", "chr:pos:A2>A1", or rsID. The same applies to locus names (PIEZO1 locus, CYBA-PIEZO1 locus). It is confusing. Please be consistent across main text (for example, rs563555492_T and PIEZO1 locus) and main figure panels. Related to this, please provide a genome browser view for PIEZO1 locus. There are references to gene symbols in the regions, such as "an intronic SNP lying between exons 4 and 5 of ZFPM1" (line 164-165), and a browser view would help contextualize the claims.

We apologise for the confusing mixture of nomenclatures and have updated the main text throughout with the gnomAD nomenclature (chr16-88716656-G-T). We have included a LocusZoom plot of the *PIEZO1* locus (see revised figure 4) for additional genomic context to help guide the reader and contextualise our claims.

4. Ancestry-specific variants vs. ancestry-specific effects. The authors should not claim ancestry-specific genetic variant effects when the lack of observed associations in non-SAS populations is likely to be explained by the lack of power due to the low allele frequency of the variant in non-SAS populations. For example, the "selective effect" in line 169 is problematic. Also, the "Heterogeneity of effect size" in line 295 should be revised.

We have amended our language through the paper to refer instead to ancestrally-heterogeneous effects, and have emphasised where relevant that most of these differences are driven purely by allele frequency.

5. Figure 5E. The reviewer has a hard time understanding the point of this analysis. The authors stratified individuals based on the genotype at rs563555492, constructed a predictive model of HbA1c using G/G carriers, and predicted HbA1c values for T/T or G/T carriers. The prediction is off, as expected because the variant rs563555492 has associations with HbA1c. What do we learn from this exercise beyond the fact that the variant is associated with HbA1c (which has already been reported elsewhere)?

The variant is strongly associated with HbA1c, and more weakly associated with glucose. It is possible that the association with HbA1c is purely mediated via an effect on glucose level. We hypothesise that this variant alters the relationship between glucose & HbA1c, i.e. the 'slope' of glucose (x axis) vs HbA1c (y axis). This analysis shows that people with a copy of the rs563555492-T allele have higher HbA1c than expected for any given level of fasting glucose, supporting the hypothesis that this variant alters the glucose-HbA1c relationship. The information this adds is that the variant is associated with HbA1c independently of its effect on glucose.

6. The corresponding hypothesis in lines 218-220 better fits in discussion. The "true" in vivo glycemic control is not observable and cannot be validated based on the results presented in the manuscript.

We have removed these lines from the results, and have weakened the claim in the discussion as follows: "Other limitations of our study regarding the phenotypes examined include the relatively small number of traits tested , the lack of clinical context for some tests (e.g. we cannot determine whether the observed lipid biomarkers were measured in a fasted state), and the static nature of these test results (e.g. we use random and fasting glucose, but do not have data on dynamic tests such as oral glucose tolerance tests). Validating the hypothesis that the *PIEZO1* missense variant chr16-88716656-G-T_T has an appreciable impact on the validity of HbA1c will therefore require recall of participants and more detailed phenotyping. Further work is required to determine whether this marker might be clinically useful, for instance in designing genotype-stratified reference ranges for HbA1c."- Line 316.

7. The "Design of reference ranges" discussion in line 304 is an overstatement. There are no results on "reference ranges" in the results.

We acknowledge this and have removed this line from the discussion. We state instead that this is an area for future work (see response to point 6).

REVIEWERS' COMMENTS

Reviewer #1 (Remarks to the Author):

The authors addressed the comments from the reviewers. I am happy to recommend this resource paper for publication. Please deposit the GWAS summary statistics to the GWAS catalog and include the accession ID in the paper during PDF proof.